# Understanding Social Reasoning in Language Models with Language Models

**Kanishk Gandhi** *    **J.-Philipp Fränken** *    **Tobias Gerstenberg**    **Noah D. Goodman**
Stanford University
{kanishk.gandhi, jphilipp}@stanford.edu

## Abstract

As Large Language Models (LLMs) become increasingly integrated into our everyday lives, understanding their ability to comprehend human mental states becomes critical for ensuring effective interactions. However, despite the recent attempts to assess the Theory-of-Mind (ToM) reasoning capabilities of LLMs, the degree to which these models can align with human ToM remains a nuanced topic of exploration. This is primarily due to two distinct challenges: (1) the presence of inconsistent results from previous evaluations, and (2) concerns surrounding the validity of existing evaluation methodologies. To address these challenges, we present a novel framework for procedurally generating evaluations *with* LLMs by populating causal templates. Using our framework, we create a new social reasoning benchmark (**BigToM**) *for* LLMs which consists of 25 controls and 5,000 model-written evaluations. We find that human participants rate the quality of our benchmark higher than previous crowd-sourced evaluations and comparable to expert-written evaluations. Using **BigToM**, we evaluate the social reasoning capabilities of a variety of LLMs and compare model performances with human performance. Our results suggest that GPT4 has ToM capabilities that mirror human inference patterns, though less reliable, while other LLMs struggle.[2]

## 1   Introduction

Humans continually try to understand what others think, want, and feel.

We try to understand what people have done and predict what they might do next by inferring their mental states. This capability, often referred to as "Theory of Mind" (ToM), is the foundation of social interaction [45, 22, 25, 10, 38]. With Large Language Models (LLMs) playing a growing role in our lives, assessing their ability to model human mental states is key for guaranteeing effective interactions. This involves evaluating the current abilities of LLMs, understanding their failure modes, and discovering ways to improve them. LLMs with ToM-like abilities could be better at teaching us, learning from us, communicating with us, collaborating with us, and understanding us [15, 20, 30, 11, 36].

Recent attempts at understanding social reasoning in LLMs have used crowd-sourced data, SocialIQA [32], data from synthetic templates, ToMi [21], or (modified) tests from psychology designed to evaluate human capabilities [e.g. 24, 42, 18, 5, 23, 41]. Sap et al. [33] used SocialIQA and ToMi to show that GPT-3 had limited social reasoning capabilities. However, their findings are challenging to interpret due to limitations in their methodology. SocialIQA has several ambiguous examples and stories that do not effectively test the desired social reasoning behaviors. In comparison, ToMi suffers from ambiguous narratives with unclear perceptual descriptions and additional confounding factors in reasoning like memory loads or tracking requirements. Moreover, both of these datasets lack control conditions making it difficult to identify precisely where models make mistakes. The results of studies with tests developed by psychologists show some signs of ToM capabilites in LLMs

---

*Equal Contribution.
[2]https://sites.google.com/view/social-reasoning-lms

37th Conference on Neural Information Processing Systems (NeurIPS 2023) Track on Datasets and Benchmarks.

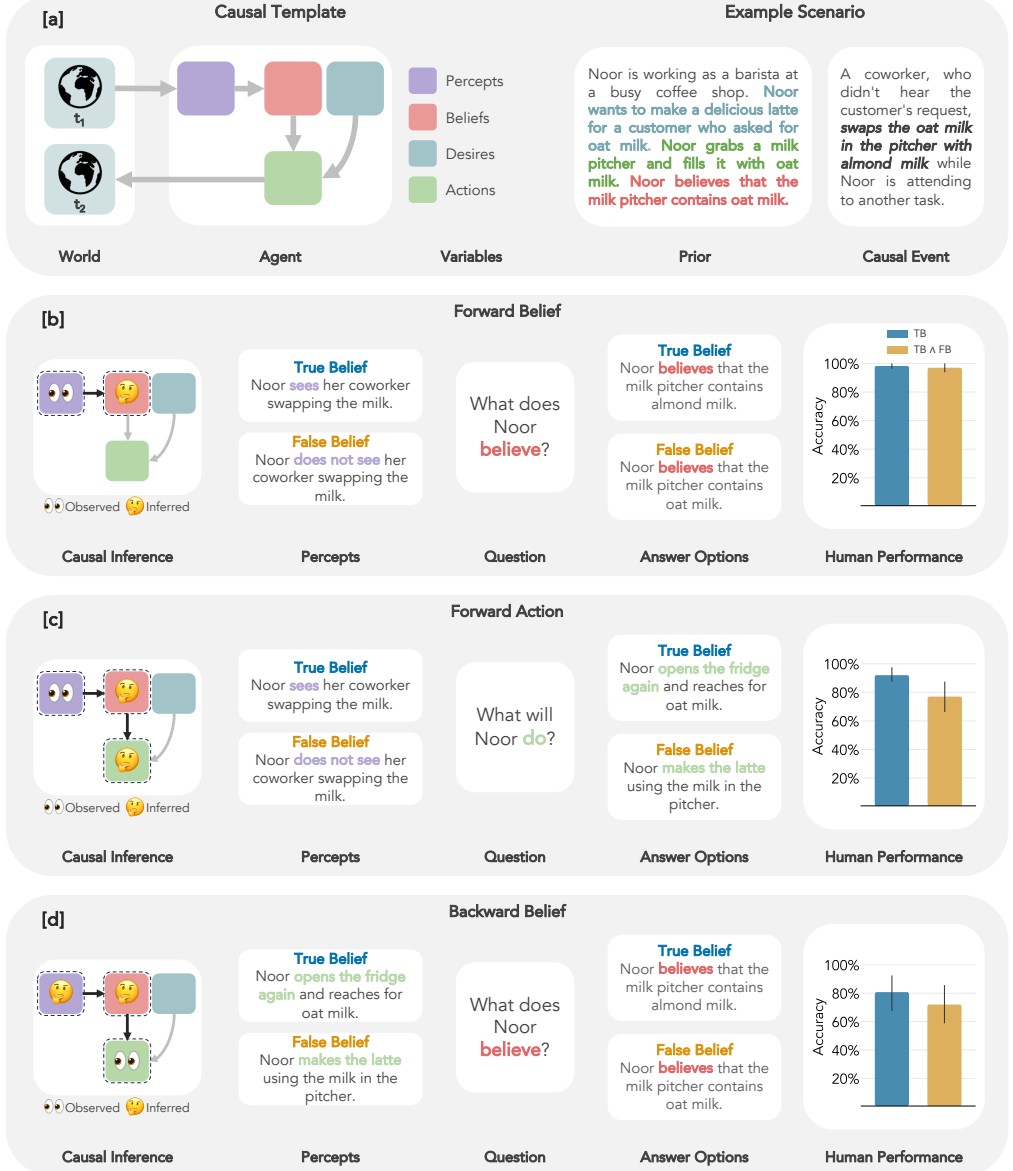

Figure 1: Illustration of our template-based Theory-of-Mind (ToM) scenarios. [a] The causal template and an example scenario including prior desires, actions, and beliefs, and a causal event that changes the state of the environment. [b] Testing *Forward Belief* inference by manipulating an agent's percepts. TB = True Belief. FB = False Belief. [c] *Forward Action* inference from an agent's percepts which requires additional inferences over unknown beliefs. [d] *Backward Belief* inference requires joint inferences over unknown percepts and beliefs from an agent's observed actions. Error bars for human performance represent 95% bootstrapped confidence intervals of the mean.

[18, 5]. However, when LLMs such as GPT-3 [4] succeed in scenarios, they often fail dramatically on trivial alterations [42, 24, 35]. Despite their careful design, concerns about the limited test set [24, 18] and potential dataset leakage from modifications to the Sally-Anne task [3] in [5, 18, 24], suggest caution in the interpretation of these results (see App. D for a detailed discussion).

To address these shortcomings, we present a novel framework for procedurally designing synthetic ToM evaluations from causal templates (Fig. 1). By representing ToM scenarios as causal graphs, we can systematically intervene on variables, generate control conditions, and probe different aspects of an LLM's ToM capabilities. More concretely, consider the scenario in Fig. 1a: Here, *"Noor"* is an agent with a desire, *"to make a latte with oat milk"*, who performed an action, *"fills it with oat milk"*, resulting in a belief, *"she believes that the pitcher has oat milk"*. Next, a *"Causal Event"*

changes the state of the environment (*"oat milk"* → *"almond milk"*). Given this setup, we can now manipulate the agent's percept to create True Belief and False Belief conditions. In the True Belief condition, the perception of the causal event is presented, *"Noor sees her coworker swapping the milk"*, and then we test a model's *forward belief* inference abilities; *"What does Noor believe is in the pitcher?"* (Fig. 1b). Moreover, we can probe more difficult inferences, such as *forward action* inferences from an agent's percepts via inferred beliefs (Fig. 1c). In addition to manipulating percepts, we can intervene on an agent's actions to examine a model's *backward belief* inferences, which is even more difficult as it requires a joint inference over unknown percepts and beliefs (Fig. 1d; §3).

We design a framework for systematic and diverse evaluations of LLMs in three steps. First, we build a causal template (an abstracted causal model) for the domain of interest, which in our case is ToM. Second, we prompt a language model to populate the variables in the template (yielding a concrete causal model). Third, we construct different evaluation conditions by combining variables from the populated causal template (Fig. 2 and §3). Our approach is a general method for generating evaluations, applicable in any domain where reasoning traces can be represented as causal graphs.

Overall, our contributions are as follows: (1) We present a framework for generating systematic evaluations from causal templates that help us understand a model's behavior, its failures and successes, through automated, controlled tests. (2) We show the effectiveness of our scalable, cost-efficient method for writing evaluations with language models by comparing its quality to crowd-sourced and expert written tests. (3) Finally, we test ToM reasoning in a variety of LLMs[3] using different prompting techniques, and compare model performances with human performance. We find that `gpt-4` shows human-like ToM inference patterns, although less reliable, while other LLMs struggle.

## 2    Related Work

**Theory-of-Mind in Humans.** Infants, arguably from 12 months of age, can attribute mental states to agents, exhibiting theory of mind reasoning [25]. A classic test to probe this reasoning is the false-belief task [3]: Sally has a doll and puts it in a basket, then leaves the room. While Sally is away, Anne takes the ball out of the basket and puts it into a box. Participants are then asked to predict what happens next: "When Sally comes back, where will she look for her ball?". To answer this question, participants need to infer Sally's beliefs, and realize that her beliefs aren't the same as theirs. Through well-planned experiments, cognitive scientists probe reasoning aspects relating to agents' desires and beliefs [22, 13, 45, 38]. These studies employ control conditions to rule out simple heuristics people might use, while searching for the cognitive mechanisms that underlie human reasoning and behavior [1, 2, 14, 47, 37, 9]. Such experiments have inspired AI researchers to design "behavioral" experiments for probing ToM in AI models [11, 36, 39, 19].

**Theory-of-Mind in Machines.** Initial attempts at building ToM representations in neural network based models [31, 30] used ToM specific tasks to train and test the models. As LLMs scaled and became better at reasoning, researchers used a small set of tests from cognitive science to claim that ToM reasoning had emerged in LLMs (GPT-3, GPT-4) [18, 5]. But, further probing using alterations and diverse scenarios showed that this reasoning was quite brittle [42, 24]. Other tests for social reasoning used crowd-sourced and synthetic evaluations to find mixed results [35, 32, 32, 21, 41]. Despite the abundance of research in this domain, we still don't understand the strengths and weaknesses of LLMs in ToM reasoning. Previous evaluations suffer from one or more of the following issues: reliance on limited evaluations designed for humans [e.g. 18, 24], insufficient control conditions [e.g. 33, 42], limited test cases [e.g. 41, 5], noisy/ambiguous crowd-sourced evaluations [e.g. 33], the risk of dataset leakage [e.g. 18, 41], confounding factors in reasoning [e.g. 21, 42] and possible overfitting of the prompting method [24] (see App. D for a detailed discussion). The goal of our work is to come up with a scalable, replicable framework to understand the reasoning behind predictions made by language models while avoiding the pitfalls that other methods fall into.

**Model-Written Evaluations.**

Advancements in aligning LLMs with instruction-tuning and RL from human feedback (RLHF) have recently shown promising results, such as the generation of a high-quality hate-speech detection dataset with GPT-3 [16, 8], red-teaming [27], and training data generation [34]. The latest work has extended this to the generation of evaluations directly [28]. Perez et al. [28] examined whether

---

[3]`LLaMa-65B`, `text-davinci-003`, `gpt-3.5-turbo`, `Claude-v1.3`, `Claude-2`, `gpt-4-0314`

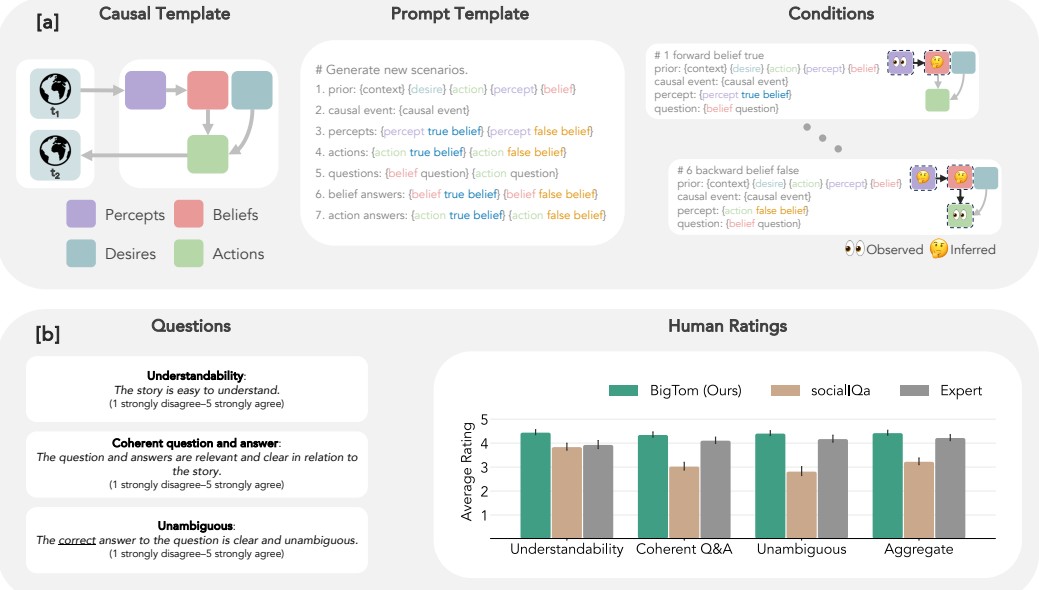

Figure 2: [a] Three-stage method for generating evaluations: Building a causal template for the domain (left). Creating a prompt template (simplified here; see Fig. 4 for the prompt) from the causal graph and populating template variables using a language model (middle). Composing test items by combining template variables (right). [b] Crowdworker ratings of our model-generated Theory-of-Mind (ToM) evaluations compared to crowd-sourced ToM evaluations and expert-written ToM evaluations. Error bars represent 95% bootstrapped confidence intervals of the mean.

generated data can serve as high-quality evaluation data with minimal errors for a variety of novel language model behaviors. These tests, while being scalable, cost-effective and easy to replicate, are still challenging to interpret as they lack structure in the generation of tests. In contrast, Dasgupta et al. [6] show how carefully designed automated tests can find specific failure modes in reasoning. Our work aims to integrate the benefits of these methods, creating a more structured approach to generating and interpreting tests, while preserving scalability, cost-effectiveness, and ease of replication.

## 3 Model-Written Evaluations with Causal Templates

**Preliminaries.** Theory of Mind is the ability to attribute mental states like beliefs, intents, desires, emotions and knowledge to oneself and others. It involves understanding that other people's mental states (latent causes) guide their actions (see Fig. 1a). In this work, we focus on the causal graph linking precepts, beliefs, desires, and actions. We want to test if models are able to perform forward and backward inference over different variables in this graph.

Our goal is to generate ToM evaluations that meet the following criteria: (1) they include control conditions to systematically assess language models' response tendencies and failure modes across different aspects of ToM, (2) they don't directly involve human-designed test items, and (3) they are diverse and scalable. By generating a diverse set of tasks, we wish to specifically target the reasoning involved in ToM inferences, while not focusing on other errors in common-sense reasoning[4]. To achieve this, we follow [28] and propose using language models to generate their own evaluations, specifically story($s$)-question($q$)-answer($a$) test items of the format of $(s_1, q_1, a_1), (s_2, q_2, a_2), ...(s_N, q_N, a_N)$ (examples are shown in Tab. 1). To generate these evaluations, we propose a novel three stage-method: (1) Building a causal template of the domain, (2) populating causal templates using language models, and (3) composing test items for a given condition by "stitching" together template variables into fluent stories (Fig. 2a).

---

[4]For example, in Shapira et al. [35], errors in understanding 'transparent access' are not ToM inference errors but errors in understanding perceptual access with transparent objects, i.e., not an error in computing what someone knows from what they see. Adding the line: "<agent> can see through transparent <object>." mitigates these errors with gpt-4.

## 3.1 Stage 1: Building a Causal Template

To build a causal template, we start by defining the variables (see Fig. 1a and Fig. 2a). The world is set up with a context and description of the agent (*"Noor is a barista [...]"*). Next, we add the initial (prior) values of the variables in the template: desire (*"Noor wants to make a latte"*), percept (*"Noor fills a pitcher with oat milk"*) and belief (*"Noor believes that the pitcher has oat milk"*). Next, a *Causal Event* changes the state of the environment (*"oat milk"* → *"almond milk"*). We can now manipulate the agent's percept of the causal event and the resulting action the agent will take. In this paper, we focus on the following inferences:

**Initial Percept to Initial Belief.** This tests if models understand that percepts (and actions) give rise to beliefs: *"Noor grabs a pitcher and fills it with oat milk"* → *"Noor believes that the milk pitcher contains oat milk"*. This is a preliminary inference that a model must perform before being able to answer more complicated questions about beliefs or actions following the causal event.

**With vs. Without Initial Belief.** We consider two version of the background (prior) scenario. In version one (*"without initial belief"*), we do not explicitly reveal the agent's initial belief (i.e. we exclude the sentence *"Noor believes that the pitcher has oat milk"*). In version two (*"with initial belief"*), we include the agent's initial belief in the scenario. Revealing the initial belief should make the inference problem easier as we can skip the inference from percept to belief. Moreover, it allows us to test whether explicitly stating the initial belief biases the answers of LLMs.

**Forward Belief.** In this condition, the model must infer the belief of the agent given the agent's percepts of the causal event (see Fig. 1b). This inference can be written as: $P(\text{Belief} \mid \text{Percept})$.

**Forward Action.** Here, the model must infer the agent's action given percepts (see Fig. 1c). Implicitly, this inference requires the model to first infer the agent's belief before predicting the agent's action given percept and desire: $\sum_{\text{Belief}} P(\text{Action} \mid \text{Percept}, \text{Desire}, \text{Belief})$.

**Backward Belief.** In this condition (Fig. 1d), the goal is to infer the agent's belief from observed actions. This is the most difficult condition as it requires joint inference over unknown beliefs and percepts from an observed action: $\sum_{\text{Percept}} \sum_{\text{Belief}} P(\text{Action} \mid \text{Desire}, \text{Percept}, \text{Belief})$.

**Additional Controls.** To control for context effects, we further include a control condition in which the "Causal Event" is replaced with a "Random Event" that does not change the state of the environment (e.g., *"A musician starts playing music while Noor is making the latte."*).

## 3.2 Stage 2: Populating Causal Templates With Language Models

Unlike previous work [28, 43], we do not directly use language models to generate individual test items. Instead, we create prompt templates (Fig. 2a, App. A) from the causal template developed in the previous section and use a language model(`gpt-4-0314` with a temperature of $0.5$ and default parameters) to fill template variables. For a given prompt, we generate $3$ new completions using $3$ few-shot examples. We constrain the model to generate exactly one sentence for a each variable in our template. Here we make an assumption that the model is good at forward prediction, coming up with plausible actions from the context, and the belief and desire of the agent (see App. C for a discussion).

## 3.3 Stage 3: Composing Test Items from Template Variables

Having generated a sentence for each variable of the template, we choose the sentences to include in the story; this varies by condition depending on the inferences we wish to test. For example, we can create a story for the *Forward Belief inference for the True Belief condition* by combining the sentences for variables `context`, `desire`, `action`, `percept`, `belief` with the sentences for `causal event` and `percept`, followed by the `belief question` and the answer options for the `true belief` and `false belief` versions (see Fig. 2a). In total, we generate 200 templates and extract 25 conditions from each template (resulting in a new benchmark consisting of 5,000 test items; see App. A for examples). For our main results with both humans and language models, we will focus on the 6 most important conditions *Forward Belief* (True Belief, False Belief), *Forward Action* (True Belief, False Belief), and *Backward Belief* (True Belief, False Belief). Results for the remaining conditions are in App. E.

Table 1: Test examples from human-written and model-written datasets.

| Dataset | Human-written or Model-written Test Example |
|---------|---------------------------------------------|
| Expert | **Story**: *"Here is a bag filled with popcorn. There is no chocolate in the bag. Yet, the label on the bag says 'chocolate' and not 'popcorn.' Sam finds the bag. She had never seen the bag before. She cannot see what is inside the bag. She reads the label."* **Question**: *"What does Sam believe the bag contains?"* **Answers**: Popcorn, Chocolate (correct answer). **Written by**: Human. **Source**: [42] |
| Expert | **Story**: *"Inside the fridge of the Airbnb host's kitchen is a can labeled as 'tuna'. However, the can actually contains sardines and no tuna. Jill, a guest who just arrived at the house, opens the fridge and sees the can. She cannot see what is inside the can, but she reads the label."* **Question**: *"What does Jill think is in the can?"* **Answers**: Sardines, Tuna (correct answer). **Written by**: Human. **Source**: [18] |
| BigToM | **Story**: *"Noor is working as a barista at a busy coffee shop. Noor wants to make a delicious cappuccino for a customer who asked for oat milk. Noor grabs a milk pitcher and fills it with oat milk. Noor believes that the milk pitcher contains oatmilk. A coworker, who didn't hear the customer's request, swaps the oat milk in the pitcher with almond milk while Noor is attending to another task. Noor sees her coworker swapping the milk."* **Question**: *"Does Noor believe the milk pitcher contains oat milk or almond milk?"* **Answers**: Noor believes the milk pitcher contains almond milk (correct answer), Noor believes the milk pitcher contains oat milk. **Written by**: GPT-4. **Condition**: Forward Belief (True Belief) With Initial Belief. |
| BigToM | **Story**: *"Mei is a pearl diver in a small coastal village in Japan. Mei wants to find a perfect pearl to give to her grandmother for her birthday. Mei spots an oyster at the bottom of the sea that looks to be the right size and age to contain a pearl. Mei believes that the oyster she spotted contains a pearl. A curious octopus opens the oyster, revealing that there is no pearl inside, and then swims away. Mei dives down to collect the oyster."* **Question**: *"Does Mei believe the oyster she spotted contains a pearl or that it is empty?"* **Answers**: Mei believes the oyster she spotted contains a pearl (correct answer), Mei believes the oyster she spotted is empty. **Written by**: GPT-4. **Condition**: Backward Belief (False Belief) With Initial Belief. |
| socialIQa | **Story**: *"Kendall persisted after being told no, and eventually had a positive effect on Lee."* **Question**: *"What will Lee want to do next?"* **Answers**: Refuse to help Kendall, Give into Kendall (correct answer), Give a punch to Kendall's face. **Written by**: Human. **Source**: [32] |
| socialIQa | **Story**: *"Lee tried to remain calm when nobody answered the phone call."* **Question**: *"What does Lee need to do before this?"* **Answers**: send a text, try again, pick up the phone (correct answer). **Written by**: Human. **Source**: [32] |

## 3.4 Quality of Generated Data

**Expert Evaluations.** Tab. 1 shows random examples from human-and model-written datasets. Our model-written examples are high-quality and closely match the pattern of examples generated by human experts. To assess the quality of our model-written dataset, we first had two experts (two authors) independently evaluate 100 model-written templates including all 25 conditions (2500 test items overall). During their evaluations, experts answered the following questions: **Question 1**: *"Does the story follow the assigned structure?"* **Answers**: 1 (Yes), 0 (No). **Question 2**: *"Does the story test the desired behavior?"* **Answers**: 1 (*"Strongly Disagree"*) to 5 (*"Strongly Agree"*). The overall percentage agreement between experts on the first question was 93.94% with mean ratings of 0.919 (95% CI: 0.859–0.970) for expert 1 and 0.960 (95% CI: 0.919–0.990) for expert 2. For the second question, average expert ratings were 4.33 (95% CI: 4.13–4.53) for expert 1 and 4.35 (95% CI: 4.18–4.52) for expert 2, both with a median rating of 5.

**Participant Evaluations.** We evaluate the quality of 200 items from BigToM with human participants[5]. Due to the large number of conditions, we gather participant ratings for the true belief and false belief versions of the forward belief condition, as exemplary versions representing the conditions. We compare participants' ratings of our model-written evaluations ("**BigToM**") with 50 random items sampled from a large-scale (38,000 items), human-written (crowd-sourced) ToM benchmark ("**socialIQa**") [32] as well as 50 random items sampled from ToM scenarios written by human researchers ("**Expert**") [7, 42, 18]. Both socialIQa and the Expert test items were selected as they have recently been used to evaluate language models' ToM capabilities [e.g. 33, 42, 18, 24, 35]. Fig. 2b shows participants' average item ratings for each dataset and question. Our model-written test items (**BigToM**) received the highest ratings for each question. Results from a Bayesian linear mixed effects regression confirmed that test-items extracted from our model-written templates were better than the crowd-sourced items, particularly in coherence and un-ambiguity, and comparable to (or better than) expert-written test items (details in §B.1).

## 4 Experiments

**Evaluating Models.** We test five large language models: `text-davinci-003`, `gpt-3.5-turbo`, `gpt-4-0314`, `claude-v1.3`, and `llama-65b-q5` (quantized)[40, 12]. All models are used with the

---

[5]Preregistration Experiment 1: https://osf.io/qxj2s. Note: We doubled the size of our participants and items upon reviewer's request. All numbers in the preregistration correspond to half of the numbers reported in this paper.

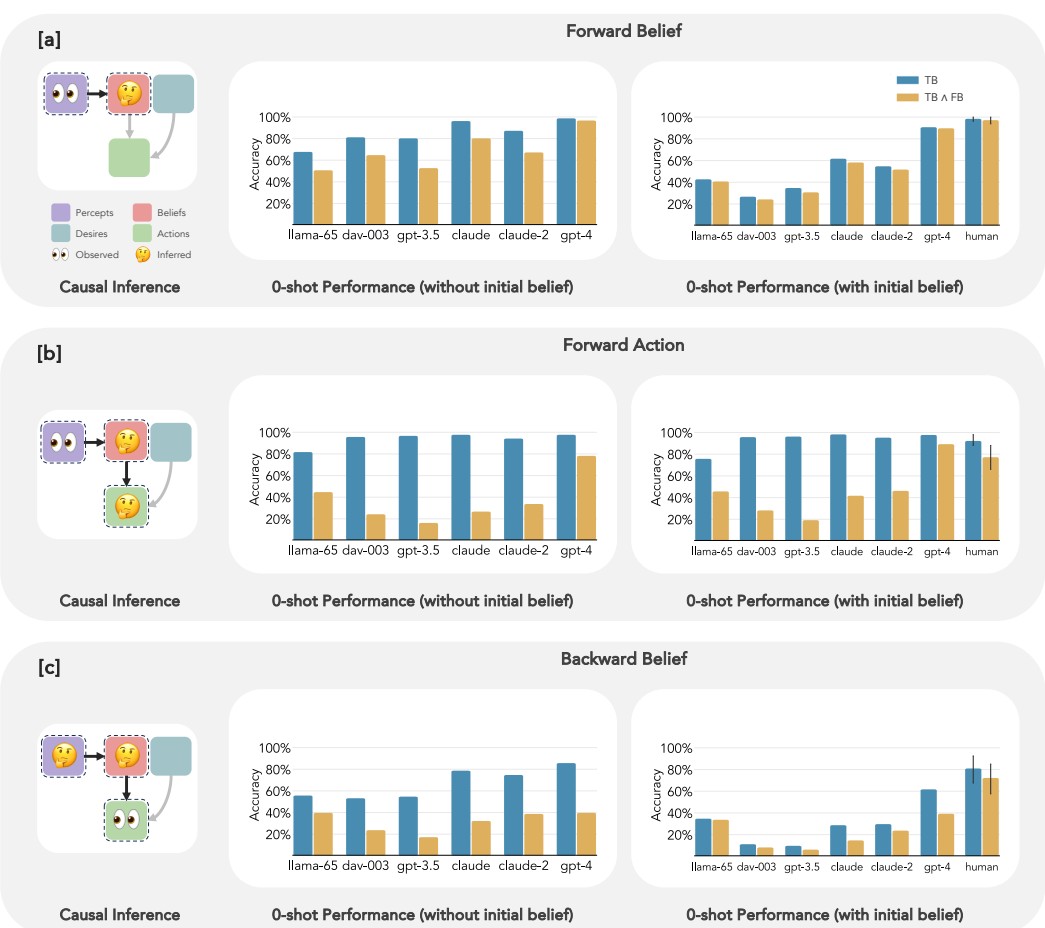

Figure 3: blueModel performance (0-shot) across conditions. [a] *Forward Belief* inferences from percepts to beliefs. TB = True Belief. FB = False Belief. [b] *Forward Action* inferences from an agent's percepts which require additional inferences over unknown beliefs. [c] *Backward Belief* inferences over unknown percepts and beliefs from an agent's observed actions. Error bars for humans represent 95% bootstrapped confidence intervals of the mean.

most deterministic setting with a temperature of 0. We test these models with four types of prompts: 0-shot, 0-shot-chain-of-thought [17], 1-shot, and 1-shot-chain-of-thought [44]. The example used for the 1-shot prompt is from the *Forward Belief - False Belief* condition, where the inference variable is the belief of the agent. The task is presented to the model in the form of a comprehension question with a story, followed by a question and two answer options. We compare models on their accuracy to answer the questions. We have released our prompts and evaluation scripts on the project page[6]. We compare models to a human baseline[7] (details in B.2).

## 4.1  Results and Discussion

The results of our investigation are detailed in Tab. 2, Tab. 7 and App. E, spanning different conditions, models, and prompts. We discuss results for the true belief and false belief conditions. Importantly, success on the false belief version of the task is evaluated *only* if the model succeeded on the true belief version, as otherwise a model might succeed on the false belief version for the wrong reasons (i.e. failing to comprehend the change in the environment rather than comprehending the change in the environment *and* understanding that the agent was not aware of this change). Therefore, we label the success on the false belief task as "TB" ∧ "False Belief".

**Initial Percept to Initial Belief.** All models are proficient at making this inference, and understand how percepts lead to the formation of beliefs (App. E to table).

---

[6]https://sites.google.com/view/social-reasoning-lms
[7]Preregistration Experiment 2: https://osf.io/zxw6m

Table 2: Performance of **GPT-4** for each method. TB = True Belief. FB = False Belief. † = without initial belief. ‡ = with initial belief.

| Condition | | Contingency | Method | | | |
|---|---|---|---|---|---|---|
| | | | *0-shot* | *0-shot-cot* | *1-shot* | *1-shot-cot* |
| *Fwd. Belief* | | TB | .99† .91‡ | .99† .99‡ | .99† .97‡ | 1.00† .97‡ |
| | | FB | .98† .99‡ | .99† .99‡ | .99† .99‡ | .99† .99‡ |
| | | TB ∧ FB | .97† .90‡ | .98† .98‡ | .97† .96‡ | .99† .96‡ |
| *Fwd. Action* | | TB | .98† .98‡ | .99† .99‡ | 1.00† 1.00‡ | 1.00† 1.00‡ |
| | | FB | .81† .92‡ | .88† .96‡ | .98† 1.00‡ | 1.00† .99‡ |
| | | TB ∧ FB | .79† .90‡ | .87† .95‡ | .98† 1.00‡ | 1.00† .99‡ |
| *Bwd. Belief* | | TB | .86† .62‡ | .84† .76‡ | .68† .57‡ | .83† .81‡ |
| | | FB | .53† .77‡ | .54† .63† | .85† .92‡ | .75† .85‡ |
| | | TB ∧ FB | .40† .40‡ | .38† .40‡ | .53† .49‡ | .58† .65‡ |

**Forward Belief Inference.** Here we test if models can track beliefs across the change in the world (Tab. 2 and Fig. 3a). Many models struggle with this, especially when an initial belief is stated (suggesting they anchor on this explicitly stated belief). `gpt-4` and, to a lesser extent, `Claude` perform better, approaching human levels.

**Forward Action Inference.** While all models are good at predicting actions when beliefs agree with the world state, most models struggle in the critical false-belief condition (Fig. 3b). `gpt-4` is the exception, exhibiting human-level performance (or even slightly better).

**Backward Belief Inference.** This represents the most challenging inference. Even humans struggle, achieving only 82% accuracy in the true belief condition and 72% in the false belief condition. We believe this is due to unavoidable uncertainty about whether the agent gained knowledge of the true world state. Models are generally far *below* chance, indicating that they reliably attribute the wrong belief, especially in false-belief situations and especially when an explicit initial belief is given (Fig. 3c). `gpt-4` is again the exception with a more human-like pattern, though not achieving human level performance 0-shot.

**Comparison of prompts.** Human participants received instructions and a demonstration example to understand the task (see App. H). Hence, a fair comparison should provide similar support to models. One-shot learning consistently enhances performance across all models and conditions. In contrast, zero-shot-chain-of-thought (CoT) prompting doesn't consistently improve performance across conditions. Introducing a one-shot CoT example does lead to consistent performance improvement across all conditions, however this performance may not be indicative of stronger ToM *per se*: mimicking the reasoning template is enough to solve our task in most cases. (Human participants were not given demonstrations of how to reason in the task.)

## 5  Discussion

In this work, we present a novel framework for measuring the capabilities of large language models (LLMs) by using abstract causal templates to automatically generate scenarios, controlling what information is provided and what must be inferred. We created a new benchmark for Theory of Mind (BigToM), allowing us to more carefully map the ToM reasoning abilities of LLMs. Our extensive control conditions aim to take into account content effects [6] and many low-level confounds. We found that many models struggle with even the most basic component of ToM, reasoning from percepts and events to beliefs, and that this is substantially affected by previously-stated beliefs. Of all the models tested, GPT-4 exhibits ToM capabilities that align closely with human inference patterns. Yet even GPT-4 was below human accuracy at the most challenging task: inferring beliefs from actions.

**Limitations.** Our evaluation methodology may appear circular at first: the model being tested plays a role in generating the test items. However, we believe that for testing inferential abilities this is not a confound. Our method constructs stories by selecting from all the available facts of a given situation and then isolates the inferential capabilities for the remaining aspects. This means that a model may be able to understand the immediate causal steps in the story while being unable to perform the required inferences being tested. Indeed, even `gpt-4` does not achieve a perfect zero-shot score at

our tests, indicating this gap between situation knowledge and inferential understanding. Further, to validate our hypothesis about circularity not being a confound, we generate an evaluation set with `claude-2`. We find that `gpt-4` gets comparable scores on an evaluation set generated by a different model, outperforming the model that created the dataset (see App. F for details).

Our method shares limitations with other model-generated evaluations (as discussed in Perez et al. [28]): the generated evaluations can be biased in the content and the contexts that are generated . While synthetic datasets generated from language models offer advantages in scale and control, they also come with inherent biases reflecting those embedded in the underlying model and training data. As large language models are trained on internet text, they inevitably pick up stereotyped associations for gender, race, and other attributes in certain contexts. This could lead to normative, stereotyped roles in different situations in the synthetic dataset. A related issue could arise from biases leading to over-generation of certain situations, effectively yielding imbalanced evaluation data. (We note this is also a problem for human-generated items!) However, language models are also steerable through detailed instructions, allowing mitigation of some biases. Careful steering might be needed during dataset generation to ensure diversity and balance along different dimensions. In domains where the models capabilities are lacking, the model will struggle to generate good evaluations. Such limitations could be resolved through shared generation with a human expert while populating the causal graph (see App. A for an example interface). The stories produced by the model at times exhibit errors in common sense, yet these instances represent a small fraction ($\sim$3%) of the overall tests generated; as language models continue to improve, we can expect these errors to reduce. Our test items tend to have syntactic similarities which might reduce the diversity of the items in our benchmark; this could perhaps be fixed by asking the model to paraphrase the generated stories.

**Future Work.** Our causal template method can be used for other domains where the effects of hidden causes or the underlying causes of effects must be inferred. These include many studied by cognitive scientists interested in the "intuitive theories" underlying human reasoning. For instance, morality, affect, and desire within social cognition, and extending to physical reasoning and more abstract reasoning such as medical diagnosis and mathematics.

In the future, testing social reasoning should move towards more realistic scenarios that are not limited to traditional ToM tests. We believe that we should focus on creating social reasoning tests or benchmarks in use-cases where LLMs are being deployed. We believe that there is a need to move towards more dynamic benchmarks for social reasoning, by creating environments where people or simulated agents (LLMs as people) interact with a language model. Such environments could also be used as a playground where the capabilities of models are not only measured, but also improved.

**Conclusion.** We have demonstrated a novel approach for assessing LLMs, and while there are limitations, we believe our findings offer a promising direction for future research in understanding and enhancing the capabilities of these powerful models. The nascent ability of LLMs to reason about mental states of people is a foundational capability for exciting use cases and problematic misuse. Systematic and broad benchmarking of these abilities is thus a pressing concern, and we believe BigToM is an important step.

## Acknowledgements

This worked was supported by the Stanford Human-Centered Artifical Intelligence (HAI) Hoffman-Yee grant, and the NSF Expeditions Grant, Award Number (FAIN) 1918771. We would like to thank the Stanford Center for Research on Foundation Models (CRFM) and Yifan Mei for the tokens and the infrastructure to test different models.

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
