| .99$^\dagger$ .91$^\ddagger$ | .99$^\dagger$ .99$^\ddagger$ | .99$^\dagger$ .97$^\ddagger$ | 1.00$^\dagger$ .97$^\ddagger$ |
| | | FB | .98$^\dagger$ .99$^\ddagger$ | .99$^\dagger$ .99$^\ddagger$ | .99$^\dagger$ .99$^\ddagger$ | .99$^\dagger$ .99$^\ddagger$ |
| | | TB ∧ FB | .97$^\dagger$ .90$^\ddagger$ | .98$^\dagger$ .98$^\ddagger$ | .97$^\dagger$ .96$^\ddagger$ | .99$^\dagger$ .96$^\ddagger$ |
| *Fwd. Action* |  | TB | .98$^\dagger$ .98$^\ddagger$ | .99$^\dagger$ .99$^\ddagger$ | 1.00$^\dagger$ 1.00$^\ddagger$ | 1.00$^\dagger$ 1.00$^\ddagger$ |
| | | FB | .81$^\dagger$ .92$^\ddagger$ | .88$^\dagger$ .96$^\ddagger$ | .98$^\dagger$ 1.00$^\ddagger$ | 1.00$^\dagger$ .99$^\ddagger$ |
| | | TB ∧ FB | .79$^\dagger$ .90$^\ddagger$ | .87$^\dagger$ .95$^\ddagger$ | .98$^\dagger$ 1.00$^\ddagger$ | 1.00$^\dagger$ .99$^\ddagger$ |
| *Bwd. Belief* |  | TB | .86$^\dagger$ .62$^\ddagger$ | .84$^\dagger$ .76$^\ddagger$ | .68$^\dagger$ .57$^\ddagger$ | .83$^\dagger$ .81$^\ddagger$ |
| | | FB | .53$^\dagger$ .77$^\ddagger$ | .54$^\dagger$ .63$^\dagger$ | .85$^\dagger$ .92$^\ddagger$ | .75$^\dagger$ .85$^\ddagger$ |
| | | TB ∧ FB | .40$^\dagger$ .40$^\ddagger$ | .38$^\dagger$ .40$^\ddagger$ | .53$^\dagger$ .49$^\ddagger$ | .58$^\dagger$ .65$^\ddagger$ |

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

state (without mentioning the agent).

Percepts:
Aware of event:  Agent perceives the event.
Not aware of event:  Agent does not perceive the event.

Actions:
Action given new state:  An action that the agent anticipates performing in response to
the object's altered state.
Action given initial state:  An action the agent might consider if the object were to
persist in its original state.

Questions:
Belief Question.
Desire Question.
Action Question.

Answers:
Belief Answers:
Aware:  Answer.
Not Aware:  Answer.
Desire Answers:
Aware:  Answer.
Not Aware:  Answer.
Action Answers:
Aware:  Answer.
Not Aware:  Answer.

Random Event.

Percepts of random event:
Agent is aware of random event.
Agent is not aware of random event.
```

Figure 4: Prompt template for generating model completions.

## A  Generating Templates

**Prompt for generating templates.** See Fig. 4 for the prompt that we use to generate the template.

**Example Template.** See Fig. 5 for an example generation.

**Interface for Generation.** When a model fails to coherently populate a template for a task, we show how are method can be extended to a semi-autonomous setting. See Fig. 7 for a sample interface for populating templates with a human-in-the-loop. It must be noted that BigToM is generated fully autonomously. This interface is only shown as an example for domains that are more difficult to generate evaluations for.

## B  Human Experiments

### B.1  Experiment 1: Human Quality Ratings

**Expert dataset.** We choose a size of 50 items as a representative sample of test items written by experts. These were sampled from three sources:

- Kosinski [18] (21/40 items): This consists of two types of false-belief ToM tasks, widely used in human studies:
  - Unexpected Contents Task [29] (20 items): In this scenario, the protagonist forms a false-belief due to a mislabeled container. The participant must infer that the protagonist incorrectly assumes that the label and its contents are aligned.

**Example model completion.**

```
Story:
Context:  Mei is a pearl diver in a small coastal village in Japan.
Desire:  Mei wants to find a perfect pearl to give to her grandmother for her birthday.
Perception cues:  Mei spots an oyster at the bottom of the sea that looks to be the
right size and age to contain a pearl.
Belief:  Mei believes that the oyster she spotted contains a pearl.
An external event:  A curious octopus opens the oyster, revealing that there is no
pearl inside, and then swims away.

Percepts of event:
Aware of event:  Mei notices the octopus opening the oyster.
Not aware of event:  Mei does not notice the octopus opening the oyster.

Actions:
Action given new state:  Mei searches for another oyster.
Action given initial state:  Mei dives down to collect the oyster.

Questions:
Belief Question:  Does Mei believe the oyster she spotted contains a pearl or that it
is empty?
Desire Question:  Does Mei want to collect the oyster she spotted, or does she want to
search for another one?
Action Question:  What will Mei do?

Answers given that Agent is Aware of the new object state:
Belief Aware:  Mei believes the oyster she spotted is empty.
Desire Aware:  Mei wants to search for another oyster.
Action Aware:  Mei will search for another oyster.

Answers given that Agent is NOT Aware of the new object state:
Belief not Aware:  Mei believes the oyster she spotted contains a pearl.
Desire not Aware:  Mei wants to collect the oyster she spotted.
Action not Aware:  Mei will dive down to collect the oyster.

Random Event:  A school of brightly colored fish swims past Mei.

Percepts of random event:
Aware of random event:  Mei sees the school of brightly colored fish.
Not aware of random event:  Mei does not notice the school of brightly colored fish
```

Figure 5: Example model completion.

- – Unexpected Transfer Task [46] (20 items): In this scenario, the protagonist witnesses an item in its original state and in the protagonist's absence, the state of this item is changed. The participant must infer that the protagonist still believes that the object is in its original state.

- Ullman [42] (9/9 items): The items in this are alterations of the tasks in Kosinski et al.

- Dodell-Feder et al. [7], Moghaddam and Honey [24]: These items are similar to the unexpected transfer task. In addition to stories about false-beliefs, this dataset also contains stories about false/outdated photographs which require the understanding of false/ outdated content.

**Procedure.** We recruited 200 human participants through Prolific [26] and asked each participant to rate 30 items (10 "Forward Belief True Belief", 10 "Forward Belief False Belief", 5 "Expert", 5 "socialIQa"), resulting in 20 independent participant ratings per item.[8]

**Results.** Fig. 2b shows participants' average item ratings for each dataset and question. Our model-written test items (**BigToM**) received the highest ratings for each question. To quantitatively compare ratings between datasets, we created an aggregate rating by taking the mean across the three likert questions for a given item and participant. We then fit a Bayesian linear mixed effects model in R included a fixed effect for the datasets and random effects for both items and participants. After fitting the model, we computed contrasts between the different datasets and found that for the contrast "BigToM-socialIQa" , the estimate was $1.152$ (95% CI: 1.066-1.244). For "BigToM-Expert", the estimate was $0.263$ (95% CI: 0.178 -0.347), and for "socialIQa-Expert", the estimate was $-0.889$ (95% CI: $-1.000$, $-0.782$). Overall, these results confirmed that test-items extracted from our model-written templates were better than the crowd-sourced items and comparable (or better) than expert-written test items. Contrasts for each dependent measure can be found in Tab. 3.

---

[8]Preregistration Experiment 1: https://osf.io/qxj2s

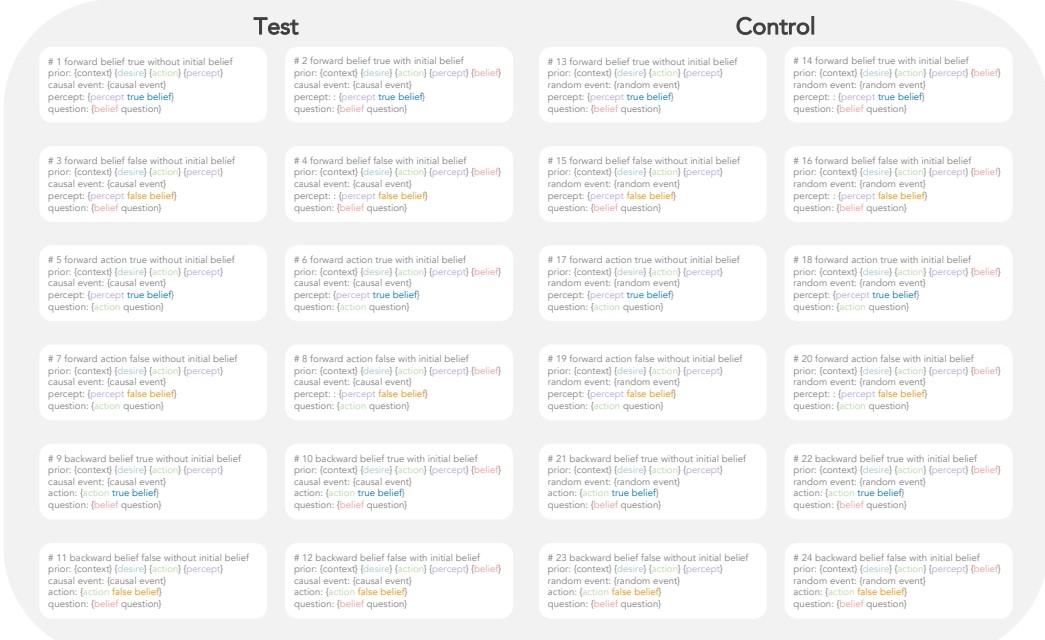

Figure 6: All 24 conditions used to evaluate models. Missing from the Figure: Condition 25 which simply included the initial percept followed by a question about the agent's initial belief (to assess whether models can infer beliefs from percepts).

Table 3: Posterior contrasts including 95% credible intervals for each dependent measure from our human quality evaluations. Aggregate = mean across the three dependent measures.

| Dependent Measure | Contrast | | |
| --- | --- | --- | --- |
| | *BigTom-socialIQa* | *BigTom-Expert* | *socialIQ-Expert* |
| Understandability | $0.503_{[0.424, 0.581]}$ | $0.248_{[0.168, 0.321]}$ | $-0.254_{[-0.346, -0.1561]}$ |
| Coherent Question & Answer | $1.406_{[1.311, 1.496]}$ | $0.203_{[0.117, 0.291]}$ | $-1.205_{[-1.315, -1.083]}$ |
| Unambiguous | $1.541_{[1.412, 1.675]}$ | $0.327_{[0.202, 0.458]}$ | $-1.214_{[-1.378, -1.057]}$ |
| Aggregate | $1.152_{[1.066, 1.244]}$ | $0.263_{[0.178, 0.347]}$ | $-0.889_{[-1.000, -0.782]}$ |

## B.2 Experiment 2: Human Performance

For the human baseline for the three main conditions (version: *"with initial belief"*), we recruited 20 participants through Prolific.[9] Participants were paid $12.05/hr.

## C Failure cases

We tried generating our templates using `text-davinci-003` and `gpt-3.5-turbo`. We found that these models were worse at following instructions and frequently made common sense errors. To mitigate this problem we tried to split the generation process into more stages: generating a context, generating initial states, beliefs and desires, generating a causal event and then stitching the story

[9]Preregistration Experiment 2: https://osf.io/zxw6m

Table 4: Source-wise quality ratings for the expert dataset.

| Data Source | Aggregate | Standard Deviation |
| --- | --- | --- |
| Dodell-Feder [7] | 4.087 | 0.907 |
| Kosinski [18] | 4.419 | 0.763 |
| Ullman [42] | 3.812 | 0.867 |

## Generate Stories

**Name of model:**

gpt-4

ID: 3

Number of stories Rated: 2

**Temperature:**

0.6

**Example Specific Instruction:**

Write a story about a farm.

Generate Story

**Context:**

Amara is a farmer in a rural Indian village. She wishes to water her crops today to ensure their growth and health. She notices that her irrigation system is functioning properly. Amara

**Perception:**

Amara sees the tree branch falling on her irrigation system.;Amara does not witness the tree branch falling on her irrigation system.

**Action:**

Amara gathers tools to repair the damaged irrigation system.;Amara waters her crops using the irrigation system.

**Belief Question:**

Does Amara believe her irrigation system is functioning properly or damaged by the fallen tree branch?

**Desire Question:**

Does Amara want to water her crops with a functioning irrigation system or with one damaged by a fallen tree branch?

**Action Question:**

What will Amara do?

**Belief Answer:**

Amara believes her irrigation system is damaged by the fallen tree branch.;Amara believes her irrigation system is functioning properly.

**Desire Answer:**

Amara wants to water her crops with a functioning irrigation system.

**Action Answer:**

Amara will gather tools to repair the damaged irrigation system.;Amara will water her crops using the irrigation system.

**Distractor:**

A group of parrots flies overhead, chattering loudly.

**Distractor Perception:**

Amara notices the parrots flying above her.;Amara does not see the parrots flying above her.

Store

Figure 7: An example interface for shared generation. This interface can be used when the model is not capable enough to populate the template by itself. It should be noted that BigToM is generated fully autonomously. Here, the user can specify the generation parameters and then edit the populated template before storing it.

Table 5: Test examples from different social reasoning benchmarks.

| Dataset | Human-written or Model-written Test Example |
|---|---|
| ADV-CFMB [42, 35] | **Story**: *"On the shelf, there is a transparent bottle. It is full of beer; there is no orange juice in it. Yet, the label on this bottle says "orange juice" and not "beer". Mark walks into the room and notices the bottle. He has never seen it before. He reads the label."* **Question**: *"What does he believe the bottle is full of?"* **Answers**: Beer (correct answer), Orange Juice . **Source**: Shapira et al. [35] |
| Kosinski [18] | **Story**: *"Inside the fridge of the Airbnb host's kitchen is a can labeled as 'tuna'. However, the can actually contains sardines and no tuna. Jill, a guest who just arrived at the house, opens the fridge and sees the can. She cannot see what is inside the can, but she reads the label."* **Question**: *"What does Jill think is in the can?"* **Answers**: Sardines, Tuna (correct answer). **Source**: Kosinski [18] |
| Moghaddam and Honey [24] | **Story**: *"The morning of the high school dance Sarah placed her high heel shoes under her dress and then went shopping. That afternoon, her sister borrowed the shoes and later put them under Sarah's bed."* **Question**: *"When Sarah gets ready, does she assume her shoes are under her dress?"* **Answers**: Yes (correct answer), No. **Source**: Dodell-Feder et al. [7] |
| ToMi [21] | **Story**: *"1 Oliver dislikes the kitchen 2 Carter entered the porch. 3 Abigail entered the porch. 4 The potato is in the green suitcase. 5 Abigail exited the porch. 6 Abigail entered the hall. 7 Carter moved the potato to the green envelope. 8 Oliver entered the hall"* **Question**: *"Where will Abigail look for the potato?"* **Answers**: green suitcase (correct answer), green envelope. **Source**: Le et al. [21] |
| socialIQa [32] | **Story**: *"Kendall persisted after being told no, and eventually had a positive effect on Lee."* **Question**: *"What will Lee want to do next?"* **Answers**: Refuse to help Kendall, Give into Kendall (correct answer), Give a punch to Kendall's face. **Source**: Sap et al. [32] |
| socialIQa [32] | **Story**: *"Lee tried to remain calm when nobody answered the phone call."* **Question**: *"What does Lee need to do before this?"* **Answers**: send a text, try again, pick up the phone (correct answer). **Source**: Sap et al. [32] |

together. Although this approach yielded improvement, the reliability of the generated content still fell short of our expectations. To further improve generations, we added verifiers (the model judging & giving feedback on the populated template) and revisers (the model revising a template based on feedback) to the pipeline. This improved the quality of generations but the number of mistakes made by the model were still high. Switching to `gpt-4` with a single generation stage gave high quality generations quite reliably. `gpt-4` occasionally fails to follow the structure of the template about 2-3% of the times. Verifying this structure is easy while parsing the template. We simply reject the templates that don't follow the assigned structure. `gpt-4` made common-sense errors (1-2%) in cases where the agent's *awareness of the change in state* in the environment was incorrect.

# D   Previous Benchmarks for ToM Reasoning in LLMs

See Tab. 5 for examples from different datasets.

**ToMi.** ToMi [21], utilizes templates to generate theory of mind queries akin to those seen in the Sally-Anne tasks. However, the scope of these tasks is fairly narrow, being limited to modifications in object locations. The perceptual access of different agents in the scene is not clearly defined. In several cases, the stories in ToMi are also ambiguous. Additionally, ToMi has several factors that potentially interfere with the accurate assessment of the theory of mind. These factors involve demands on memory and tracking; the questions posed are extensive, necessitating the simultaneous tracking of multiple locations and agents. Finally, a lack of control conditions makes some evaluations with this dataset difficult to interpret.

**SocialIQA.** These tests have been crowd-sourced and tend to be quite noisy with very ambiguous answers. Several questions in the dataset don't test the desired behaviors. The lack of any structure to the dataset makes evaluations with this benchmark difficult to interpret.

**Bubeck et al. [5], Kosinski [18]** Tests in these works have a high similarity with other false-belief tasks that have been used with humans. Success on these could merely be the result of dataset leakage. Trivial alterations of the tasks, as shown by Ullman [42] lead to failure. Finally, the number of test examples in these benchmarks is quite low, making it difficult to systematically test capabilities.

**Moghaddam and Honey [24]** Tests in Moghaddam and Honey [24] (from Dodell-Feder et al. [7]) face limitations similar to Bubeck et al. [5], Kosinski [18]. Moghaddam and Honey [24] show that one-shot CoT leads to a perfect score on their set of tests. These results are difficult to interpret as all questions require a similar reasoning structure to solve them. So, success with the one-shot CoT example may reduce the generalization gap to an extent that theory of mind capabilities aren't actually being tested.

**Ullman [42], Shapira et al. [35]** Ullman [42] introduce alterations of tests from Kosinski [18] to show how models make mistakes with theory of mind inferences. Based on this dataset, Shapira et al.

Table 6: Five additional example test items from our model-written dataset.

| Dataset | Model-written Test Example |
|---|---|
| BigToM | **Story**: *"Rashid is a hiker who has set out to climb a challenging mountain trail. His goal is to reach the summit before sunset to enjoy the panoramic view. Rashid observes that the weather is clear and sunny, perfect for hiking. Rashid believes that the weather conditions are ideal for his hike. As Rashid continues his ascent, a dense fog suddenly rolls in, reducing visibility on the trail. Rashid does not notice the dense fog."* **Question**: *"What will Rashid do?"* **Answers**: Rashid will confidently hike towards the summit (correct answer), Rashid will consider waiting for the fog to clear before continuing his hike. **Written by**: GPT-4. **Condition**: Forward Action (False Belief) With Initial Belief. |
| BigToM | **Story**: *"Aarav is a landscape architect working on a new public park. He wants to plant a row of trees that will provide shade and beauty to the park's visitors. Aarav sees that the saplings he has ordered appear to be strong and healthy. However, an unexpected heatwave hits the area, causing the saplings to wilt and weaken. Aarav does not notice the heatwave's effects on the saplings."* **Question**: *"Does Aarav believe the saplings are strong and healthy or wilted and weakened?"* **Answers**: Aarav believes the saplings are strong and healthy (correct answer), Aarav believes the saplings are wilted and weakened. **Written by**: GPT-4. **Condition**: Forward Belief (False Belief) Without Initial Belief. |
| BigToM | **Story**: *"Naveen is a professional baker who runs a small bakery in a bustling city. Today, he plans to bake a large batch of chocolate chip cookies for an upcoming event. Naveen notices that the oven is preheated to the correct temperature. He believes that the oven is ready for baking. Naveen listens to the street musician's performance. Naveen starts baking the chocolate chip cookies."* **Question**: *"Does Naveen believe the oven is preheated to the correct temperature or affected by the power outage and cooled down?"* **Answers**: Naveen believes the oven is preheated to the correct temperature (correct answer), Naveen believes the oven is affected by the power outage and has cooled down. **Written by**: GPT-4. **Condition**: Backward Belief Control (False Belief) With Initial Belief. |
| BigToM | **Story**: *"Dalia is a professional photographer, setting up her equipment for a wedding photoshoot. She wants to capture stunning photos with her high-quality camera. Dalia sees that the camera lens appears clean and free of smudges. Dalia believes that the camera lens is clean and free of smudges. While Dalia is away, a mischievous child plays with the camera, leaving fingerprints all over the lens. Dalia notices the fingerprints on the camera lens."* **Question**: *"What will Dalia do?"* **Answers**: Dalia will carefully clean the lens before starting the photoshoot (correct answer), Dalia will begin the photoshoot with her clean camera lens. **Written by**: GPT-4. **Condition**: Forward Action (True Belief) With Initial Belief. |
| BigToM | **Story**: *"Azola is a teacher at a small school in rural China. She wants to surprise her students with a new book for the school library. Azola got a notification from Amazon stating that the package containing the book was left at her doorstep. Azola believes the book she ordered has arrived in the package at her doorstep. A gust of wind blows the package off her doorstep, and a neighbor replaces it with a different package containing a hand-knit scarf. Azola retrieves the original package with the book."* **Question**: *"Does Azola believe the package contains the book she ordered or a hand-knit scarf?"* **Answers**: Azola believes the package contains a hand-knit scarf (correct answer), Azola believes the package contains the book she ordered. **Written by**: GPT-4. **Condition**: Backward Belief (True Belief) With Initial Belief. |

[35] introduce a new benchmark, ADVersarial CommonSense with False-Belief (Adv-CSFB) with a larger number of test examples to test for theory-of-mind. Models, including `gpt-4`, fair poorly on this benchmark. We believe that failures on this benchmark are due to a lack of information about perceptual access, i.e., failures in understanding how some situations change perceptual access, and not failure in inferences related to theory of mind. For example, adding the line "<agent> can see through transparent <object>" leads to success on tasks with transparent access with `gpt-4`. With problems relating to uninformative labels, where the label is in a different language, adding the line "<agent> cannot read <differnt language>." leads to success on the task.

Figure 8: Instructions shown to participants in Experiment 1.

Figure 9: Prompts used to evaluate the model. For chat based models, the instructions are presented as a system message and the questions are presented as a user message.

# E   Evaluating models

**Resources.** The resources used for our experiments include the APIs associated with the models; in particular the OpenAI API and the Anthropic API. The llama model was evaluated on an internal cluster using a single NVIDIA A40 GPU.

**Prompts.** See Fig. 9 for the 0-shot prompt, Fig. 10 for the 0-shot CoT prompt, Fig. 11 for the 1-shot prompt, Fig. 12 for the 1-shot CoT prompt.

Figure 10: Prompts used to evaluate the model. For chat based models, the instructions are presented as a system message and the questions are presented as a user message.

Figure 11: Prompts used to evaluate the model. For chat based models, the instructions are presented as a system message and the questions are presented as a user message.

Figure 12: Prompts used to evaluate the model. For chat based models, the instructions are presented as a system message and the questions are presented as a user message.

**Results of models.** See Tab. 7 for the results of all models across different conditions.

**Results on Controls.** See Tab. 8 for results of the model on the control conditions.

Table 7: Model performance for each method. TB = True Belief. FB = False Belief. $^{\dagger}$ = without initial belief. $^{\ddagger}$ = with initial belief.

| Model | Condition | Contingency | Method | | | |
|---|---|---|---|---|---|---|
| | | | *0-shot* | *0-shot-cot* | *1-shot* | *1-shot-cot* |
| llama-65 | *Fwd. Belief* | TB | .68$^\dagger$ .43$^\ddagger$ | .40$^\dagger$ .34$^\ddagger$ | .82$^\dagger$ .62$^\ddagger$ | .42$^\dagger$ .31$^\dagger$ |
| | | FB | .62$^\dagger$ .72$^\ddagger$ | .77$^\dagger$ .74$^\ddagger$ | .82$^\dagger$ .89$^\ddagger$ | .75$^\dagger$ .75$^\dagger$ |
| | | TB ∧ FB | .51$^\dagger$ .41$^\ddagger$ | .34$^\dagger$ .31$^\ddagger$ | .71$^\dagger$ .60$^\ddagger$ | .36$^\dagger$ .28$^\dagger$ |
| | *Fwd. Action* | TB | .82$^\dagger$ .76$^\ddagger$ | .31$^\dagger$ .31$^\dagger$ | .95$^\dagger$ .93$^\ddagger$ | .34$^\dagger$ .27$^\dagger$ |
| | | FB | .47$^\dagger$ .52$^\ddagger$ | .19$^\dagger$ .43$^\dagger$ | .50$^\dagger$ .58$^\ddagger$ | .20$^\dagger$ .39$^\dagger$ |
| | | TB ∧ FB | .45$^\dagger$ .46$^\ddagger$ | .10$^\dagger$ .20$^\dagger$ | .49$^\dagger$ .55$^\ddagger$ | .10$^\dagger$ .15$^\dagger$ |
| | *Bwd. Belief* | TB | .56$^\dagger$ .35$^\ddagger$ | .25$^\dagger$ .18$^\ddagger$ | .53$^\dagger$ .31$^\ddagger$ | .25$^\dagger$ .19$^\dagger$ |
| | | FB | .53$^\dagger$ .73$^\ddagger$ | .69$^\dagger$ .75$^\ddagger$ | .69$^\dagger$ .84$^\ddagger$ | .66$^\dagger$ .71$^\dagger$ |
| | | TB ∧ FB | .40$^\dagger$ .34$^\ddagger$ | .21$^\dagger$ .18$^\ddagger$ | .38$^\dagger$ .26$^\ddagger$ | .21$^\dagger$ .19$^\dagger$ |
| dav-003 | *Fwd. Belief* | TB | .82$^\dagger$ .27$^\ddagger$ | .84$^\dagger$ .42$^\ddagger$ | .61$^\dagger$ .25$^\ddagger$ | .76$^\dagger$ .35$^\ddagger$ |
| | | FB | .82$^\dagger$ .98$^\ddagger$ | .85$^\dagger$ .98$^\ddagger$ | .99$^\dagger$ .99$^\ddagger$ | .86$^\dagger$ .97$^\ddagger$ |
| | | TB ∧ FB | .65$^\dagger$ .25$^\ddagger$ | .69$^\dagger$ .34$^\ddagger$ | .60$^\dagger$ .24$^\ddagger$ | .63$^\dagger$ .32$^\ddagger$ |
| | *Fwd. Action* | TB | .96$^\dagger$ .96$^\ddagger$ | .97$^\dagger$ .99$^\ddagger$ | .99$^\dagger$ .99$^\ddagger$ | .98$^\dagger$ .98$^\ddagger$ |
| | | FB | .27$^\dagger$ .31$^\ddagger$ | .22$^\dagger$ .30$^\ddagger$ | .56$^\dagger$ .66$^\ddagger$ | .19$^\dagger$ .23$^\ddagger$ |
| | | TB ∧ FB | .25$^\dagger$ .29$^\ddagger$ | .21$^\dagger$ .29$^\ddagger$ | .56$^\dagger$ .65$^\ddagger$ | .18$^\dagger$ .22$^\ddagger$ |
| | *Bwd. Belief* | TB | .54$^\dagger$ .12$^\ddagger$ | .65$^\dagger$ .22$^\ddagger$ | .40$^\dagger$ .10$^\ddagger$ | .54$^\dagger$ .19$^\ddagger$ |
| | | FB | .59$^\dagger$ .96$^\ddagger$ | .57$^\dagger$ .92$^\ddagger$ | .84$^\dagger$ .97$^\ddagger$ | .65$^\dagger$ .92$^\ddagger$ |
| | | TB ∧ FB | .24$^\dagger$ .09$^\ddagger$ | .30$^\dagger$ .16$^\ddagger$ | .28$^\dagger$ .07$^\ddagger$ | .25$^\dagger$ .14$^\ddagger$ |
| gpt-3.5 | *Fwd. Belief* | TB | .81$^\dagger$ .35$^\ddagger$ | .90$^\dagger$ .54$^\ddagger$ | .83$^\dagger$ .41$^\ddagger$ | .95$^\dagger$ .97$^\ddagger$ |
| | | FB | .69$^\dagger$ .95$^\ddagger$ | .54$^\dagger$ .86$^\ddagger$ | .88$^\dagger$ .97$^\ddagger$ | .97$^\dagger$ .97$^\ddagger$ |
| | | TB ∧ FB | .53$^\dagger$ .31$^\ddagger$ | .48$^\dagger$ .45$^\ddagger$ | .73$^\dagger$ .38$^\ddagger$ | .93$^\dagger$ .85$^\ddagger$ |
| | *Fwd. Action* | TB | .97$^\dagger$ .97$^\ddagger$ | .96$^\dagger$ .97$^\ddagger$ | .96$^\dagger$ .96$^\ddagger$ | .96$^\dagger$ .97$^\ddagger$ |
| | | FB | .19$^\dagger$ .22$^\ddagger$ | .11$^\dagger$ .16$^\ddagger$ | .59$^\dagger$ .73$^\ddagger$ | .72$^\dagger$ .83$^\ddagger$ |
| | | TB ∧ FB | .17$^\dagger$ .20$^\ddagger$ | .10$^\dagger$ .14$^\ddagger$ | .55$^\dagger$ .69$^\ddagger$ | .69$^\dagger$ .81$^\ddagger$ |
| | *Bwd. Belief* | TB | .55$^\dagger$ .10$^\ddagger$ | .62$^\dagger$ .19$^\ddagger$ | .55$^\dagger$ .23$^\ddagger$ | .86$^\dagger$ .70$^\ddagger$ |
| | | FB | .45$^\dagger$ .92$^\ddagger$ | .30$^\dagger$ .77$^\ddagger$ | .65$^\dagger$ .87$^\ddagger$ | .51$^\dagger$ .76$^\ddagger$ |
| | | TB ∧ FB | .18$^\dagger$ .07$^\ddagger$ | .09$^\dagger$ .06$^\ddagger$ | .28$^\dagger$ .15$^\ddagger$ | .38$^\dagger$ .49$^\ddagger$ |
| claude | *Fwd. Belief* | TB | .97$^\dagger$ .62$^\ddagger$ | .90$^\dagger$ .61$^\ddagger$ | .92$^\dagger$ .82$^\ddagger$ | 1.00$^\dagger$ .98$^\ddagger$ |
| | | FB | .82$^\dagger$ .97$^\ddagger$ | .88$^\dagger$ .98$^\ddagger$ | .98$^\dagger$ .99$^\ddagger$ | .99$^\dagger$ .99$^\ddagger$ |
| | | TB ∧ FB | .81$^\dagger$ .59$^\ddagger$ | .80$^\dagger$ .59$^\ddagger$ | .91$^\dagger$ .82$^\ddagger$ | .99$^\dagger$ .97$^\ddagger$ |
| | *Fwd. Action* | TB | .98$^\dagger$ .99$^\ddagger$ | .97$^\dagger$ .95$^\ddagger$ | .96$^\dagger$ .96$^\ddagger$ | 1.00$^\dagger$ 1.00$^\ddagger$ |
| | | FB | .28$^\dagger$ .43$^\ddagger$ | .43$^\dagger$ .49$^\ddagger$ | .92$^\dagger$ .98$^\ddagger$ | .98$^\dagger$ 1.00$^\ddagger$ |
| | | TB ∧ FB | .27$^\dagger$ .42$^\ddagger$ | .41$^\dagger$ .46$^\ddagger$ | .88$^\dagger$ .93$^\ddagger$ | .97$^\dagger$ 1.00$^\ddagger$ |
| | *Bwd. Belief* | TB | .79$^\dagger$ .29$^\ddagger$ | .74$^\dagger$ .29$^\ddagger$ | .59$^\dagger$ .22$^\ddagger$ | .79$^\dagger$ .73$^\ddagger$ |
| | | FB | .48$^\dagger$ .80$^\ddagger$ | .55$^\dagger$ .83$^\ddagger$ | .89$^\dagger$ .96$^\ddagger$ | .76$^\dagger$ .81$^\ddagger$ |
| | | TB ∧ FB | .33$^\dagger$ .15$^\ddagger$ | .39$^\dagger$ .22$^\ddagger$ | .48$^\dagger$ .20$^\ddagger$ | .56$^\dagger$ .55$^\ddagger$ |
| claude-2 | *Fwd. Belief* | TB | .88$^\dagger$ .55$^\ddagger$ | .93$^\dagger$ .87$^\ddagger$ | .99$^\dagger$ .67$^\ddagger$ | 1.00$^\dagger$ .95$^\ddagger$ |
| | | FB | .75$^\dagger$ .94$^\ddagger$ | .95$^\dagger$ .99$^\ddagger$ | .95$^\dagger$ .98$^\ddagger$ | .99$^\dagger$ .99$^\ddagger$ |
| | | TB ∧ FB | .68$^\dagger$ .52$^\ddagger$ | .89$^\dagger$ .61$^\ddagger$ | .84$^\dagger$ .67$^\ddagger$ | .99$^\dagger$ .95$^\ddagger$ |
| | *Fwd. Action* | TB | .95$^\dagger$ .96$^\ddagger$ | .96$^\dagger$ .95$^\ddagger$ | .89$^\dagger$ .91$^\ddagger$ | .99$^\dagger$ 1.00$^\ddagger$ |
| | | FB | .36$^\dagger$ .49$^\ddagger$ | .51$^\dagger$ .70$^\ddagger$ | .90$^\dagger$ 1.00$^\ddagger$ | .97$^\dagger$ .96$^\ddagger$ |
| | | TB ∧ FB | .34$^\dagger$ .47$^\ddagger$ | .49$^\dagger$ .65$^\ddagger$ | .83$^\dagger$ 1.00$^\ddagger$ | .95$^\dagger$ .96$^\ddagger$ |
| | *Bwd. Belief* | TB | .75$^\dagger$ .30$^\ddagger$ | .82$^\dagger$ .34$^\ddagger$ | .45$^\dagger$ .22$^\ddagger$ | .61$^\dagger$ .39$^\ddagger$ |
| | | FB | .50$^\dagger$ .82$^\ddagger$ | .60$^\dagger$ .88$^\ddagger$ | .89$^\dagger$ .96$^\ddagger$ | .92$^\dagger$ .96$^\ddagger$ |
| | | TB ∧ FB | .39$^\dagger$ .24$^\ddagger$ | .43$^\dagger$ .26$^\ddagger$ | .41$^\dagger$ .21$^\ddagger$ | .53$^\dagger$ .35$^\ddagger$ |
| gpt-4 | *Fwd. Belief* | TB | .99$^\dagger$ .91$^\ddagger$ | .99$^\dagger$ .99$^\ddagger$ | .99$^\dagger$ .97$^\ddagger$ | 1.00$^\dagger$ .97$^\ddagger$ |
| | | FB | .98$^\dagger$ .99$^\ddagger$ | .99$^\dagger$ .99$^\ddagger$ | .99$^\dagger$ .99$^\ddagger$ | .99$^\dagger$ .99$^\ddagger$ |
| | | TB ∧ FB | .97$^\dagger$ .90$^\ddagger$ | .98$^\dagger$ .98$^\ddagger$ | .97$^\dagger$ .96$^\ddagger$ | .99$^\dagger$ .96$^\ddagger$ |
| | *Fwd. Action* | TB | .98$^\dagger$ .98$^\ddagger$ | .99$^\dagger$ .99$^\ddagger$ | 1.00$^\dagger$ 1.00$^\ddagger$ | 1.00$^\dagger$ 1.00$^\ddagger$ |
| | | FB | .81$^\dagger$ .92$^\ddagger$ | .88$^\dagger$ .96$^\ddagger$ | .98$^\dagger$ 1.00$^\ddagger$ | 1.00$^\dagger$ .99$^\ddagger$ |
| | | TB ∧ FB | .79$^\dagger$ .90$^\ddagger$ | .87$^\dagger$ .95$^\ddagger$ | .98$^\dagger$ 1.00$^\ddagger$ | 1.00$^\dagger$ .99$^\ddagger$ |
| | *Bwd. Belief* | TB | .86$^\dagger$ .62$^\ddagger$ | .84$^\dagger$ .76$^\ddagger$ | .68$^\dagger$ .57$^\ddagger$ | .83$^\dagger$ .81$^\ddagger$ |
| | | FB | .53$^\dagger$ .77$^\ddagger$ | .54$^\dagger$ .63$^\dagger$ | .85$^\dagger$ .92$^\ddagger$ | .75$^\dagger$ .85$^\ddagger$ |
| | | TB ∧ FB | .40$^\dagger$ .40$^\ddagger$ | .38$^\dagger$ .40$^\ddagger$ | .53$^\dagger$ .49$^\ddagger$ | .58$^\dagger$ .65$^\ddagger$ |

## F  Addressing Concerns of Circularity

We generate an alternate evaluation set using Claude-2 with 30 populated templates and 750 conditions called BigToM-Claude. We evaluate the source models, Claude-2 and gpt-4 on this evaluation set. The performance of gpt-4-0314 on BigToM-Claude is similar to that of BigToM-gpt4 (see Tab. 11). This shows that our tests are consistent regardless of the source that populates the template. Moreover, Claude-2, despite generating BigToM-Claude performs worse than gpt-4 on this dataset. With this validation set generated by Claude, we are able to show that the abilities tested during evaluation are distinct from the ones that are used while populating the template. To further validate this, we ran a logistic regression with data source (BigToM-gpt4, BigToM-Claude) and model type (gpt-4, Claude-2) as predictors and response (correct, incorrect) as dependent variable. To match the smaller size of BigToM-Claude, we randomly sampled a subset of BigToM-gpt4. The results of this analysis confirmed that there was no significant interaction ($b$ = -0.13, $p$ = 0.620), while there was a significant main effect of model type, with gpt-4 performing better across conditions than Claude-2 ($b$ = 1.07, $p$ < .001). Model performances for each data source are shown in Tab. 10.

## G  Diversity in BigToM

We analyze the diversity of the dataset by asking GPT-4 to annotate the 200 templates along 3 different dimensions, types of changeable states, types of mechanisms of change, types of medium of perceptions. These are generic labels provided by GPT-4 for each of the stories (see Tab. 12).

We show some examples of these categories here:

- Examples of mechanisms of change: rainfall, monkeys, power surge, leak, toddler, fog, cat, vandals, malfunction
- Examples of mediums of perception: tastes, sees, feels, hears, smells, reads, discovers/notices (ambiguous)
- Examples of changeable states: water level, type of ingredient, temperature, visibility, dryness, cleanliness, color, tuning, sharpness

These descriptive statistics show that our generated items have substantially diverse content. In contrast, expert written datasets are very limited in their number and hence diversity due to the time and effort involved in generating the evaluations. For example most expert items only involve a change in location of the object (changeable state: location), which is usually done by an agent (mechanism of change: person) and is perceived by looking at the agent changing the object.

## H  Participant Instructions and Pay

**Personally Identifiable Info and IRB**. Participants completed a consent page before the start of each experiment. Participants were made aware of potential risks and links to the Stanford IRB approvals were provided on the consent page.

**Duration and Pay**. For Experiment 1, participants were paid $12.02/hr. The median completion time of Experiment 1 was 21 minutes and 55 seconds. The total cost for Experiment 1 was $512.00. For Experiment 2, participants were paid $12.05/hr. The median completion time of Experiment 2 was 21 minutes and 22 seconds. The total cost for Experiment 2 was $114.40.

## I  Licenses

We release the dataset with an MIT License, see https://sites.google.com/view/social-reasoning-lms.

## J  Extension to Second Order Beliefs

In Fig. 15, we show how our template based method can be extended to testing the inference of second order beliefs.

Table 8: Model performance for controls. TB = True Belief. FB = False Belief. † = without initial belief. ‡ = with initial belief.

| Model | Condition | Contingency | Method 0-shot |
|---|---|---|---|
| **llama-65** | Fwd. Belief | TB | .81† .94‡ |
| | | FB | .86† .96‡ |
| | | TB ∧ FB | .79† .92† |
| | Fwd. Action | TB | .53† .68‡ |
| | | FB | .56† .72‡ |
| | | TB ∧ FB | .50† .66† |
| | Bwd. Belief | TB | .90† .96‡ |
| | | FB | .90† .96‡ |
| | | TB ∧ FB | .90† .96† |
| **dav-003** | Fwd. Belief | TB | .98† .99‡ |
| | | FB | .98† .99‡ |
| | | TB ∧ FB | .98† .99† |
| | Fwd. Action | TB | .88† .91‡ |
| | | FB | .93† .94‡ |
| | | TB ∧ FB | .88† .90† |
| | Bwd. Belief | TB | .98† .99‡ |
| | | FB | .98† .99‡ |
| | | TB ∧ FB | .98† .99† |
| **gpt-3.5** | Fwd. Belief | TB | .99† .99‡ |
| | | FB | .98† .99‡ |
| | | TB ∧ FB | .98† .99† |
| | Fwd. Action | TB | .83† .85‡ |
| | | FB | .82† .87‡ |
| | | TB ∧ FB | .77† .79† |
| | Bwd. Belief | TB | .99† .99‡ |
| | | FB | .99† .99‡ |
| | | TB ∧ FB | .99† .99† |
| **claude** | Fwd. Belief | TB | .99† .99‡ |
| | | FB | .99† .99‡ |
| | | TB ∧ FB | .99† .99† |
| | Fwd. Action | TB | .88† .93‡ |
| | | FB | .86† .93‡ |
| | | TB ∧ FB | .83† .89† |
| | Bwd. Belief | TB | .99† .99‡ |
| | | FB | .99† .99‡ |
| | | TB ∧ FB | .99† .99† |
| **gpt-4** | Fwd. Belief | TB | .99† .99‡ |
| | | FB | .99† .99‡ |
| | | TB ∧ FB | .99† .99† |
| | Fwd. Action | TB | .98† .95‡ |
| | | FB | .99† .98‡ |
| | | TB ∧ FB | .98† .94† |
| | Bwd. Belief | TB | .99† .99‡ |
| | | FB | .99† .99‡ |
| | | TB ∧ FB | .99† .99† |

Table 9: Model Performance Initial Percept to Initial Belief.

| Model | Method |
|---|---|
| | *0-shot* |
| **llama-65** | .92 |
| **dav-003** | .98 |
| **gpt-3.5** | .96 |
| **claude** | .98 |
| **gpt-4** | .99 |

Table 10: Claude-2 vs. gpt-4 accuracy on BigTom-Claude and BigTom-gpt4.

| Model Name | Data Source | |
|---|---|---|
| | *BigTom-Claude* | *BigTom-gpt4* |
| Claude-2 | 0.675 | 0.708 |
| gpt-4 | 0.858 | 0.861 |

Table 11: GPT-4's and Claude-2's performance for each method on 30 test items generated by Claude-2. TB = True Belief. FB = False Belief. † = without initial belief. ‡ = with initial belief.

| Model | Condition | Contingency | Method | | | |
|---|---|---|---|---|---|---|
| | | | *0-shot* | *0-shot-cot* | *1-shot* | *1-shot-cot* |
| **claude-2** | *Fwd. Belief* | TB | 1.00† .60‡ | .97† .80‡ | .60† .34‡ | .90† .77‡ |
| | | FB | .67† .97‡ | .94† .97‡ | .97† 1.00‡ | 1.00† 1.00‡ |
| | | TB ∧ FB | .67† .57‡ | .90† .77‡ | .60† .34‡ | .90† .77‡ |
| | *Fwd. Action* | TB | .94† .94‡ | .90† .97‡ | .84† .80‡ | .97† 1.00‡ |
| | | FB | .34† .54‡ | .54† .78‡ | .84† .77‡ | .97† 1.00‡ |
| | | TB ∧ FB | .30† .47‡ | .47† .74‡ | .74† .64‡ | .94† 1.00‡ |
| | *Bwd. Belief* | TB | .84† .40‡ | .97† .54‡ | .37† .30‡ | .70† .50‡ |
| | | FB | .17† .74‡ | .37† .84‡ | .77† .87‡ | .97† 1.00‡ |
| | | TB ∧ FB | .10† .24‡ | .34† .47‡ | .34† .30‡ | .70† .50‡ |
| **gpt-4** | *Fwd. Belief* | TB | .97† .94‡ | .97† .87‡ | .97† .94‡ | .97† .94‡ |
| | | FB | 1.00† 1.00‡ | 1.00† 1.00‡ | 1.00† 1.00‡ | 1.00† 1.00‡ |
| | | TB ∧ FB | .97† .94‡ | .97† .87‡ | .97† .94‡ | .97† .94‡ |
| | *Fwd. Action* | TB | .94† .97‡ | .97† .97‡ | .94† .97‡ | .94† 1.00‡ |
| | | FB | .64† .97‡ | .87† .97‡ | 1.00† 1.00‡ | 1.00† 1.00‡ |
| | | TB ∧ FB | .57† .94‡ | .84† .94‡ | .94† .97‡ | .94† 1.00‡ |
| | *Bwd. Belief* | TB | 1.00† .80‡ | .94† .84‡ | .84† .70‡ | .90† .74‡ |
| | | FB | .37† .74‡ | .50† .70† | .97† 1.00‡ | .97† .97‡ |
| | | TB ∧ FB | .37† .54‡ | .44† .54‡ | .84† .70‡ | .87† .70‡ |

Table 12: Measures of diversity of the evaluations in our dataset.

| Changeable States | Mechanisms of Change | Mediums of perception |
|---|---|---|
| 154 | 76 | 7 |

Figure 13: Instructions shown to participants in Experiment 1.

Figure 14: Instructions shown to participants in Experiment 2.

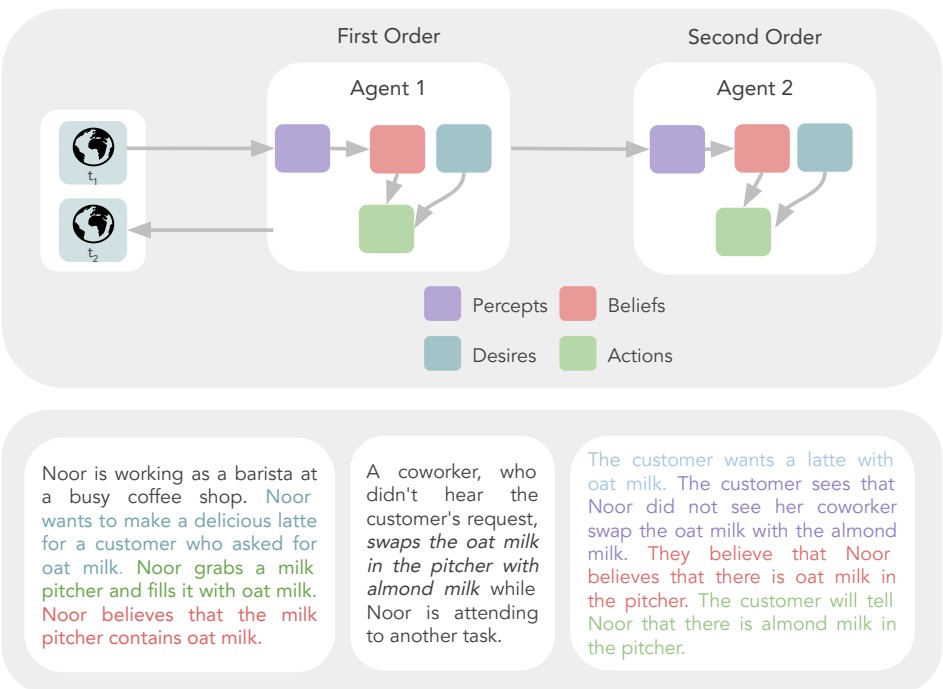

Figure 15: An example template for testing the capabilities of inferring second order beliefs.