# OpenReview forum: "Understanding Social Reasoning in Language Models with Language Models"
_NeurIPS.cc/2023/Track/Datasets_and_Benchmarks — NeurIPS 2023 Datasets and Benchmarks Spotlight_

### Official Review · Reviewer_URe5 · 2023-07-20
**BigToM, a procedurally generated dataset and full framework for measuring the social reasoning of LLMs.**

**Rating:** 9
**Confidence:** 4
**Clarity:** The paper and repository are exceptio…

**Strengths:**

The importance of evaluating the social reasoning of LLMs is well motivated.

The dataset is large and diverse, consisting of 200 generated templates applied in 25 different control-condition combinations to produce 5000 total evaluations.


**Additional Feedback:**

Despite the high quality of the writing, many obvious misspellings remain. For instance, see lines 110, 121, 163.

**Correctness:**

The BigToM evaluations are rated by human participants to be of consistently higher quality than other datasets, and are even comparable to expert-written evaluations.

The tests are methodologically thorough, covering 0 and 1-shot cases, both with and without chain-of-thought prompting.


**Documentation:**

The repository is excellent, providing full installation instructions, detailed steps for generating the dataset, plus code to evaluate the models.

**Ethics:**

No concerns.

**Limitations:**

No concerns.

**Opportunities For Improvement:**

I find little in this work to complain about! The discussion section covers several interesting lines of future work that could go even further.

**Relation To Prior Work:**

The paper presents a comprehensive summary of prior works evaluating ToM reasoning in LLMs, and explains their weaknesses, including methodology, effectiveness, ambiguity, confounding factors, and lack of controls.

**Summary And Contributions:**

The BigToM dataset is generated by GPT-4 from causal templates designed to represent theory-of-mind (ToM) scenarios as causal graphs, in order to enable interventions, controls, and probes.

Five LLMs were evaluated:  text-davinci-003, gpt-3.5-turbo, gpt-4-0314, claude-v1.3, and llama-65b-q5.

The results show that GPT-4 approaches human capabilities in these types of social reasoning, while other LLM lag behind.

---

> ### Author Response · Authors · 2023-08-22
> **Response to Reviewer URe5**
>
> We are grateful for the reviewer's positive reading of the paper. We are glad that they found the paper ‘well motivated’ and the dataset ‘large and diverse’. We are happy about their comments on the clarity of our paper and code, ‘paper and repository are exceptionally clear and well-written.’
>
> We thank the reviewer for pointing out the typos in our draft; we have corrected them in the revision. Please also see the general response above indicating other changes that we make.  Let us know if you have any questions or concerns about the work we could address during the discussion period.

---

> > ### Comment · Reviewer_URe5 · 2023-08-26
> > **Response**
> >
> > Great to see the new experiments and improvements to the paper!

---

### Official Review · Reviewer_m3qS · 2023-07-21
**Review of BigToM**

**Rating:** 8
**Confidence:** 3

**Strengths:**

The authors systematically design belief-desire situations and provide inferences on probable beliefs and actions. I welcome this kind of ToM variant as a timely new benchmark because of the emerging importance of mind state reasoning in the LLM era. The dataset construction framework is useful because it enables scalable future research in ToM field, and the storyline of each data sample is challenging compared with previous works.

**Additional Feedback:**

I have a question to the authors.
- Do the authors perform iteration on each experiment? If not, is it because every model is set to zero temperature?

**Clarity:**

This paper is clearly written and details are sufficiently specified in the supplementary material.

**Correctness:**

The proposed dataset is constructed by following causal templates representing several kinds of inference situations.

**Documentation:**

Every consideration is satisfied.

**Limitations:**

It would be great if the authors specify the limitation of the proposed dataset and its potential bias in a separated section.

**Opportunities For Improvement:**

It would be great if second-order belief situations are adopted together, but the proposed causal graph structure would make it easier to extend the work in the future.

**Relation To Prior Work:**

The authors describe previous works in supplementary material and clearly specify the contribution of the proposed dataset.

**Summary And Contributions:**

This paper proposes template-based controllable framework simulating Theory-of-Mind (ToM). The proposed generation framework can scale up many ToM situations and enable language models evaluate the reasoning capability easily. Specifically, it handles the language model's inference ability in forward belief, forward action, and backward belief by conditioning and combining causal events.

---

> ### Author Response · Authors · 2023-08-22
> **Response to Reviewer m3qS**
>
> We thank the reviewer for their thoughtful comments on our work. We are happy that they found our benchmark ‘timely’ and our method ‘useful’ with its scalability and cost-efficiency. We address their concerns here:
>
> > It would be great if second-order belief situations are adopted together, but the proposed causal graph structure would make it easier to extend the work in the future.
>
> Our work was an initial effort in understanding inferences in situations with first-order beliefs. As the reviewer notes, our casual template can naturally extend to second-order belief situations. We attach a potential causal graph for this in the Appendix J of our paper, though we feel it is beyond the scope of this initial paper to generate, validate, and evaluate items in this setting.
>
> > It would be great if the authors specify the limitation of the proposed dataset and its potential bias in a separated section.
>
> We split the discussion to include a separate subsection on the limitations of the dataset. We also expand on the discussion of the potential bias in the dataset, adding this paragraph to the limitations:
>
> While synthetic datasets generated from language models offer advantages in scale and control, they also come with inherent biases reflecting those embedded in the underlying model and training data. As large language models are trained on internet text, they inevitably pick up stereotyped associations for gender, race, and other attributes in certain contexts. This could lead to normative, stereotyped roles in different situations in the synthetic dataset. A related issue could arise from biases leading to over-generation of certain situations, effectively yielding imbalanced evaluation data. (We note this is also a problem for human-generated items!) However, language models are also steerable through detailed instructions, allowing mitigation of some biases. Careful steering might be needed during dataset generation to ensure diversity and balance along different dimensions.
>
> We thank Reviewer m3qS for their feedback. Please also see the general response above indicating other changes that we make. Let us know if you have any questions or concerns about the work that we could address in the discussion period.

---

> > ### Comment · Reviewer_m3qS · 2023-08-29
> >
> > Great! Thank you for responding my comments. I have no further questions.

---

### Official Review · Reviewer_rvRQ · 2023-07-21

**Rating:** 6
**Confidence:** 2
**Correctness:** The paper is correct to my knowledge.
**Clarity:** The paper is well-written.

**Strengths:**

* A framework proposal for generating systematic evaluations from causal templates
* Cost-efficient and scalable approach
* Acceptable variety of LLMs were selected and evaluated with different prompting
* The framework takes into account low-level confounds and content effects

**Additional Feedback:**

Please see comments above

**Documentation:**

There are a few potential false claims in the author checklist. For instance, there isn't any mention regarding resource usage in the paper nor appendix (to my best knowledge); however, it's marked as [YES] in the checklist. Also, the study does not contain training process. I recommend mentioning the section number where the corresponding question in the checklist is addressed.

**Ethics:**

Not to my knowledge

**Limitations:**

In limitations section (starting line 260), a solution for lack of good evaluation generation is "through shared generation with a human expert". Isn't relying on human expert for evaluation generation a limitation itself?

**Opportunities For Improvement:**

* The use of more LLMs for evaluation in the future (low priority)
* more detailed explanation of the dataset used in this study
* dataset size used for BigToM seems small (based on the data files on Github); clarification regarding this matter may be a good idea


**Relation To Prior Work:**

Yes it is.

**Summary And Contributions:**

This paper proposes a new framework for generating evaluations using LLMs with causal templates. It addresses two main challenges from previous approaches: inconsistent results from previous evaluations and concerns about the validity of existing evaluation methods. This work creates a social reasoning benchmark called BigToM. Using this benchmark, the social reasoning abilities of various LLMs are evaluated and are compared to human performance.

---

> ### Author Response · Authors · 2023-08-22
> **Response to Reviewer rvRQ**
>
> We thank the reviewer for their thoughtful comments on our work. We are glad that they found the paper to be ‘well-written’ and the approach to be ‘cost-efficient and scalable’.
>
> > dataset size used for BigToM seems small (based on the data files on Github); clarification regarding this matter may be a good idea
>
> We have 200 populated templates (present in `data/bigtom.csv`) from which we generate 25 conditions for each template. This gives us a total of 5000 evaluations to test the models with.
>
> > a solution for lack of good evaluation generation is "through shared generation with a human expert". Isn't relying on human expert for evaluation generation a limitation itself?
>
> The dataset that we generate is **completely autonomous**, without human intervention. For problems where language models **aren’t yet good enough** to generate evaluations, we suggest a shared generation interface. While this method of shared generation is limiting compared to the fully autonomous condition, it is much more efficient compared to fully relying on the human expert to generate evaluations. This allows tapping into human expertise while retaining the efficiency of language model generation.
>
> > there isn't any mention regarding resource usage in the paper nor appendix (to my best knowledge); however, it's marked as [YES] in the checklist. Also, the study does not contain training process. I recommend mentioning the section number where the corresponding question in the checklist is addressed.
>
> We have updated the checklist in the paper to include section numbers with each response. We have also updated the response for the training process to ‘N/A’. The resources used for our experiments include the APIs associated with the models; in particular the OpenAI API and the Anthropic API. We have also added that the evaluation with the llama model was done on an internal cluster using a single NVIDIA A40 GPU.
>
> > more detailed explanation of the dataset
>
> We add additional analysis of the generated dataset here. We look at the types of objects, the changes that these objects undergo and the types of contexts generated here:
>
> We analyze the diversity of the dataset by asking GPT-4 to annotate the 200 templates along 3 different dimensions, types of changeable states, types of mechanisms of change, types of medium of perceptions. These are generic labels provided by GPT-4 for each of the stories:
>
> |Changeable States|Mechanisms of Change|Mediums of perception|
> |-----|-----|-----|
> |154|76|7|
>
>
> We show some examples of these categories here:
> - Examples of changeable states: water level, type of ingredient, temperature, visibility, dryness, cleanliness, color, tuning, sharpness
> - Examples of mechanisms of change: rainfall, monkeys, power surge, leak, toddler, fog, cat, vandals, malfunction
> - Examples of mediums of perception: tastes, sees, feels, hears, smells, reads, discovers/notices (ambiguous)
>
> We have added this analysis to the Appendix Section G.
>
> We thank Reviewer rvRQ for their feedback. Please also see the general response above indicating other changes that we make. Let us know if you have any other questions or concerns about the work that we could address.

---

### Official Review · Reviewer_KFdV · 2023-07-21
**Useful methods but I have concerns about certain claims**

**Rating:** 6
**Confidence:** 4
**Correctness:** I have some concerns.
**Clarity:** The paper is well written overall.

**Strengths:**

The proposed templated method solves a concrete problem of creating a “clean” eval set and can be useful for evaluating and hillclimb for ToM  and social reasoning tasks.

The proposed templated method for synthetic data generation is also generalizable.

**Additional Feedback:**

N/A

**Documentation:**

Yes

**Limitations:**

No.

See opportunities of improvement above.

**Opportunities For Improvement:**

The biggest issue of the work is generating the eval task, and evaluating it on the same models (and additional models). This can be circular. And the submission addresses this by mentioning “Our evaluation methodology may appear circular at first: the model being tested plays a role in generating the test items.”
There are a variety of ways to address this. E.g., run a validation study to use different models to generate the eval set, and evaluate it on a different set of models and see if it changes the conclusions drawn. Or simply address it as a limitation and leave for future work.

However, given the framing of the paper in the very beginning is the shortcomings in current eval methodology, and in particular, “(2) concerns surrounding the validity of existing evaluation methodologies.”, I would hold the authors to a higher bar here.

I find the claim “However, we believe that for testing inferential abilities this is not a confound and is even a virtue.” not convincing, and even distracting. I would suggest removing these types of claims and just be upfront about the limitations and need for future validation.


**Relation To Prior Work:**

Yes

**Summary And Contributions:**

The paper presents a framework for generating evaluation dataset for ToM and reports eval results across several models. The proposed approach falls under synthetic data/eval set generation and is interesting. The proposed dataset and approach can add value to the social reasoning / ToM reasoning literature. But I do have concerns over the circular eval method, and the fact that the submission claims it as a “virtue”.

---

> ### Author Response · Authors · 2023-08-22
> **Response to Reviewer KFdV**
>
> We thank the reviewer for their thoughtful comments on our work. We are glad that they think our method ‘add(s) value’ to the literature, and that our method ‘solves a concrete problem of creating a “clean” eval set’. We address their concerns here:
>
> > have concerns over the circular eval method…evaluating it on the same models (and additional models). This can be circular...run a validation study to use different models to generate the eval set, and evaluate it on a different set of models…claim “However, we believe that for testing inferential abilities this is not a confound and is even a virtue.” not convincing, ...removing these types of claims and just be upfront about the limitations and need for future validation.
>
> To address the reviewer’s concerns regarding the circularity of the method, we generate a new dataset using Claude-2 and evaluate GPT-4 using it. We add the results to the Appendix Section F.
>
> We generate an alternate evaluation set using Claude-2 with 750 items called BigToM-Claude. We evaluate the source model, Claude-2 and gpt-4 on this evaluation set. The performance of gpt-4-0314 on BigToM-Claude is similar to that of BigToM-gpt4. This shows that our tests are consistent regardless of the source that populates the template. Moreover, Claude-2, despite generating BigToM-Claude performs worse than gpt-4 on this dataset. With this validation set generated by Claude, we are able to show that the abilities tested during evaluation are distinct from the ones used to populate the template.
>
> Based on the reviewer’s feedback, we change our paragraph on circularity to:
>
> Our evaluation methodology may appear circular at first: the model being tested plays a role in generating the test items. However, we believe that for testing inferential abilities this is not a confound ~~and even a virtue. That is, in order to test whether a model can reason about latent beliefs for a given common-sense situation, we must first know that the model understands the (non-mental) situation. Using the model to fill our causal template makes this more likely; we then further tested this in control conditions.~~ Our method constructs stories by selecting from all the available facts of a given situation and then isolates the inferential capabilities for the remaining aspects. This means that a model may be able to understand the immediate causal steps in the story while being unable to perform the required inferences being tested. Indeed, even gpt-4 does not achieve a perfect zero-shot score at our tests, indicating this gap between situation knowledge and inferential understanding. Further, to validate our hypothesis about circularity not being a confound, we generate an evaluation set with claude-2. We find that gpt-4 gets comparable scores on an evaluation set generated by a different model, outperforming the model that created the dataset.
>
>
> We thank Reviewer KFdV for their feedback. We hope that the reviewer might consider raising their score in the light of these responses. Please also see the general response above indicating other changes that we make. Let us know if you have any other questions or concerns about the work that we could address.

---

### Official Review · Reviewer_ffAP · 2023-07-21
**Interesting work, but one potential concern that arises is the possibility of circularity in the methodology.**

**Rating:** 6
**Confidence:** 4

**Strengths:**

A significant advantage of their created dataset is the incorporation of control conditions, setting it apart from previous datasets used for evaluating LLMs.
The experiment conducted to evaluate the generated benchmark involves a comprehensive analysis that tests 25 conditions for each scenario with various LLMs and compares them with human participants’ performance.

**Additional Feedback:**

n/a

**Clarity:**

The discussion on previous benchmarks (Appendix D) is well-written and effectively motivates the need for a new benchmark.
Regarding the causal graph variables, in section 3.1, it is implied that the values of the variables are manually created, whereas the prompt template in Figure 2.a and section 3.2, suggest that the model generates the values. Also, the prompt template in Figure 2.a is not the same template that is provided in Figure 4. To address these inconsistencies, the authors should consider having a single figure that presents the exact prompt, including all instructions, alongside what is generated by the model. This will help clear any ambiguity.
In Figure 7, it is shown that users can add specific instructions, but the prompt template in Figure 4 does not demonstrate how these instructions are incorporated into the prompt.
Regarding the model used for generating the 200 stories, although it is not explicitly mentioned, based on Appendix C, it appears that GPT4 was used. To provide clarity, please explicitly state which model was used, and if the samples were created in multiple stages using different methods, explain this process thoroughly as it constitutes a core part of the paper.
Regarding the tables and figures where the performance is reported, could you please clarify the performance metric?
How are the correct answers to the generated questions from the stories determined?
Minor comment: Page 9 → “We have demonstrated a novel approach for assessing LLMs”
	Minor typo: Appendix A →  See Fig. 4 for the prompt

**Correctness:**

The insufficient sample size for data quality evaluation (Refer to the second paragraph in Opportunities for improvement)

**Documentation:**

There is a GitHub link in the paper where the data and code for generating the dataset and running the experiments are provided

**Limitations:**

The authors point out the limitations regarding the generated content and that it can be biased.

**Opportunities For Improvement:**

It is notable that the authors included human participants in rating their benchmark alongside stimuli from previous studies. However, the data quality evaluation relies on 25 samples from each of the previous datasets. The SocialIQA contains 38,000 items. This small sample size is insufficient to yield meaningful results about the dataset (power analysis and sample size determination are suggested). As a result, the data quality evaluation result might not be reliable.
The other dataset, “Expert”, is not clearly explained. While three previous papers are cited as sources of the expert samples (Dodell-Feder et al.; Kosinsk;, and Ullman), it is only mentioned that 25 items were randomly selected and combined to create the "Expert" dataset. A more detailed explanation of each dataset, along with the rationale for combining them, would be appreciated, especially considering the potential differences in the distributions of the 3 datasets.
The proposed framework is claimed to generate diverse examples. However, the term "diverse set of tasks" remains ambiguous. It would be beneficial to address these questions in the publication: How exactly is diversity measured and ensured in the generated examples, and how does this benchmark compare to others in terms of diversity?  Additionally, the mention in the discussion section that the test items have syntactic similarities, which could reduce diversity, contrasts with the claim of diversity.

**Relation To Prior Work:**

The paper identifies and discusses the limitations of previous benchmarks. In response to these limitations, the authors introduce their framework for creating a new benchmark.

**Summary And Contributions:**

The paper proposes a framework to systematically create a social reasoning benchmark using LLMs and argues that the benchmark is superior to crowd-sourced and expert-written benchmarks. The authors use the benchmark to evaluate Theory of Mind (TOM) capabilities of LLMs, compare them to that of humans, and conclude that GPT4 has comparable performance with human inference, but other LLMs struggle with social reasoning.

---

> ### Author Response · Authors · 2023-08-22
> **Response to Reviewer ffAP (1/2)**
>
> We thank the reviewer for their thoughtful comments on our work. We are glad that they liked our ‘incorporation of control conditions’ that set ‘it apart from previous datasets used for evaluating LLMs’. We address their concerns here:
>
> > data quality evaluation relies on 25 samples from each of the previous datasets. The SocialIQA contains 38,000 items. This small sample size is insufficient to yield meaningful results about the dataset. As a result, the data quality evaluation result might not be reliable …
>
> We have now **doubled** the size of the data sampled from each dataset, resulting in 50 items each from socialIQA and expert datasets, and 200 from BigTom overall. We had to use a subsample of items from SocialIQA to reduce the cost associated with human annotation.
> We argue that this sample and the ratings are representative of the full dataset as:
>
> - The findings from our initial 50% sample remain consistent (see Table 3) with the doubled dataset.
> - Our analysis using Bayesian regression conveys through credible intervals that the true value of the ratings for the datasets lies within the reported intervals with 95% confidence (see Appendix section B.1 for details).
>
> > “Expert”, is not clearly explained… it is only mentioned that 25 items were randomly selected and combined to create the "Expert" dataset. A more detailed explanation of each dataset, along with the rationale for combining them, would be appreciated, especially considering the potential differences in the distributions of the 3 datasets.
>
> We describe the composition of the “Expert” dataset here:
> - Kosinski (21/40 items): This consists of two types of false-belief ToM tasks, widely used in human studies:
>     - Unexpected Contents Task (20 items) [1]:  In this scenario, the protagonist forms a false-belief due to a mislabeled container. The participant must infer that the protagonist incorrectly assumes that the label and its contents are aligned.
>     - Unexpected Transfer Task (20 items) [2]: In this scenario, the protagonist witnesses an item in its original state and in the protagonist's absence, the state of this item is changed. The participant must infer that the protagonist still believes that the object is in its original state.
> - Ullman (9/9 items): The items in this are alterations of the tasks in Kosinski et al.
> - Dodell-Feder et al. / Moghadam and Honey (20/20 items): These items are similar to the unexpected transfer task. In addition to stories about false-beliefs, this dataset also contains stories about false/outdated photographs which require the understanding of false/ outdated content.
>
> While reporting our results, we combine these datasets since we wanted to compare how tests developed by different sources (synthetic, crowdsourced, expert) compared with each other.
>
> To further elaborate on our results, we report source-wise statistics here:
>
> |Sources||Aggregate Quality| Standard Deviation|
> |-----|-----|------|
> |Kosinski|4.087|0.907|
> |Ullman|4.419|0.763|
> |Dodell-Feder|3.812|0.867|
>
> We have clarified the composition of our “expert” dataset and their description in section 3.4 and the appendix section B, including the above statistics.
>
> [1] J. Perner, S. R. Leekam, H. Wimmer, Three-year-olds’ difficulty with false belief: The case for a conceptual deficit. British Journal of Developmental Psychology (1987), doi:10.1111/j.2044-835x.1987.tb01048.x. 41.
> [2] H. Wimmer, J. Perner, Beliefs about beliefs: Representation and constraining function of wrong beliefs in young children’s understanding of deception. Cognition (1983), doi:10.1016/0010-0277(83)90004-5.
>
> *Continued (1/2)*

---

> > ### Author Response · Authors · 2023-08-22
> > **Response to Reviewer ffAP (2/2)**
> >
> > >  "diverse set of tasks" remains ambiguous
> >
> > We analyze the diversity of the dataset by asking GPT-4 to annotate the 200 templates along 3 different dimensions, types of changeable states, types of mechanisms of change, types of medium of perceptions. These are generic labels provided by GPT-4 for each of the stories:
> >
> > |Changeable States|Mechanisms of Change|Mediums of perception|
> > |-----|-----|-----|
> > |154|76|7|
> >
> >
> > We show some examples of these categories here:
> >
> > - Examples of changeable states: water level, type of ingredient, temperature, visibility, dryness, cleanliness, color, tuning, sharpness
> > - Examples of mechanisms of change: rainfall, monkeys, power surge, leak, toddler, fog, cat, vandals, malfunction
> > - Examples of mediums of perception: tastes, sees, feels, hears, smells, reads, discovers/notices (ambiguous)
> >
> > These descriptive statistics show that our generated items have substantially diverse content. In contrast, expert written datasets are very limited in their number and hence diversity due to the time and effort involved in generating the evaluations. For example most expert items only involve a change in location of the object (changeable state: location), which is usually done by an agent (mechanism of change: person) and is perceived by looking at the agent changing the object. Lastly, we emphasize semantic variation in the benchmark since large language models are adept at adapting to syntactic changes.
> >
> >
> > > in section 3.1, values of the variables are manually created, …  Figure 2.a and section 3.2, suggest that the model generates values…the prompt template in Figure 2.a is not the same template that is provided in Figure 4.
> >
> > In section 3.1, that is the first stage of our method, the user defines the causal template and the variables that they care about but not the values that these variables have. For theory of mind, these include “perception”,“beliefs”, “desires”, “actions” and the states of the world. In Stage 2 (section 3.2), the LLM populates these variables with values. We will clarify this to make the distinction clear.
> >
> > The prompt template shown in Figure 2.a is a simplification of the actual prompt shown in Figure 4. We use a simplified version in Figure 2.a so that it is easier to convey the essence of our method to the reader. We will clarify this in the caption of Figure 2a, including a reference to Figure 4.
> >
> > > In Figure 7, it is shown that users can add specific instructions, but the prompt template in Figure 4 does not demonstrate how these instructions are incorporated into the prompt.
> >
> > The dataset in the paper is generated **fully autonomously** using a command-line interface. The interface shown in Figure 7 showcases how our method can easily be extended to use cases where:
> > - a user wants more supervision over the data generation process,
> > - or where the model is not perfect at populating the templates and the user wants to interactively fill in the template *with* the model
> >
> > We have revised the figure caption and associated section to reflect this.
> >
> > > To provide clarity, please explicitly state which model was used, and if the samples were created in multiple stages using different methods, explain this process thoroughly as it constitutes a core part of the paper.
> >
> > As specified in Section 3.2, we use `gpt-4-0314` with a temperature of 0.5 and default parameters to populate the causal template (currently specified in the footnote; we have moved this to the main text). The method has 3 stages:
> > - Stage 1 (Sec 3.1): The user builds a causal template, defining relevant variables that structures based on the domain.
> > - Stage 2 (Sec 3.2): The LLM populates the template, filling in values for the variables.
> > - Stage 3 (Sec 3.3): The populated variables from the template are stitched together (by concatenating the different strings).
> >
> > Please let us know if there are aspects of this process that could be further clarified!
> >
> > > could you please clarify the performance metric? How are the correct answers to the generated questions from the stories determined?
> >
> > The evaluations are presented to LLMs in the form of comprehension questions with two options for answering the question. So, each evaluation condition consists of a story, a question, and two answer options. These are generated from the populated causal template as described in section 3.2. The correct answer option is thus known as part of the populated causal template. To compute answer accuracy we simply look at the ratio of correct answers in the responses of a model.
> >
> > We thank Reviewer ffAP for their constructive feedback. Please also see the general response above indicating other changes that we make. Let us know if you have any other questions or concerns about the work that we could address.

---

> > ### Comment · Reviewer_ffAP · 2023-08-30
> >
> > Thank you for the comprehensive response, and additional experiments with doubled samples sizes

---

### Author Response · Authors · 2023-08-22
**General Response**

General Response
It’s wonderful to see the positive response around our work – from Reviewer m3qS’s note, “​I welcome this kind of ToM variant as a timely new benchmark” to Reviewer KFdV’s opinion that our paper “solves a concrete problem of creating a ‘clean’ eval set” We’re grateful that the reviewers found our paper “exceptionally clear and well-written.”

We are also appreciative of their constructive feedback, questions, and concerns; to fully address everything, we have **added new experimental results, updated the main text and supplemental with additional discussion, and added clarity/additional arguments to answer specific concerns.** All revisions are shown in blue in the updated paper (deletions have been strike-throughed). Specifically, we have added:

**New Experiments/ Analyses:**

- **Dataset Quality Measurements  (sec 3.4 and appendix B.1)** [Reviewer ffAP]:

    - We have doubled the number of items for SocialIQA, BigTom, and Expert datasets evaluated by human participants.
    - We have provided posterior contrasts and their corresponding 95% credible intervals for all dependent measures in Table 3. Our analysis utilizes a Bayesian linear mixed-effects model to compute contrasts and associated credible intervals. Unlike frequentist regression, Bayesian regression does not yield p-values directly. Instead, credible intervals convey the statistical uncertainty surrounding parameter estimates, such as contrasts. Specifically, for our results, there's a 95% probability that the true value of the contrasts lies within the reported intervals in Table 3. If these intervals exclude zero, it suggests that the observed effects are statistically ‘significant’, analogous to ‘a p-value of less than 0.05’ in a frequentist model.
    - We now report source-wise average ratings and standard deviations for the expert dataset in Table 4

- **Addressing Circularity (appendix F)**  [Reviewer KFdV]: We have generated an alternate evaluation set using Claude-2 with 750 items called BigToM-Claude. We evaluated the source model, Claude-2 and gpt-4 on this evaluation set. The performance of gpt-4-0314 on BigToM-Claude is similar to that of BigToM-gpt4. This shows that our tests are consistent regardless of the source that populates the template. Moreover, Claude-2, despite generating BigToM-Claude performs worse than gpt-4 on this dataset. With this validation set generated by Claude, we are able to show that the abilities tested during evaluation are distinct from the ones that are used while populating the template. To further validate this, we ran a logistic regression with data source (BigToM-gpt4, BigToM-Claude) and model type (gpt-4, Claude-2) as predictors and response (correct, incorrect) as dependent variable. To match the smaller size of BigToM-Claude, we randomly sampled a subset of BigToM-gpt4. The results of this analysis confirmed that there was no significant interaction, while there was a significant main effect of model type, with gpt-4 performing better on BigToM-gpt4 and BigToM-Claude as compared to Claude-2. Regression results are provided in appendix F and model performances for each data source are shown in Table 10.
- **Diversity in the dataset  (appendix G)** [Reviewer ffAP, rvRQ] : We have added measures of diversity along three dimensions for our tasks: type of object (the object that changes state in the story), cause of change (what causes a change in the state of the object), and type of percept (how does the agent perceive the change in state). We show how our evaluations are more diverse compared to the expert written ones.
- **Claude-2 Results (Figure 3, appendix E, Table 7)**

**Summary of Major Revisions:**

- **Updated Limitations (section 5)** [Reviewer m3qS]: We have split our discussion to have a separate section for limitations. We have expanded on the biases that might be present in the dataset due to the use of a language model.
- **Updated Checklist** [Reviewer rvRQ ]: We have updated the checklist with references to the sections where the questions have been addressed.

We hope that our revisions and more detailed individual responses below cover our reviewers’ outstanding concerns. If there are any remaining concerns, we look forward to addressing them over the rest of the discussion period! Thank you all!

---

### Decision · Program_Chairs · 2023-09-22

**Decision:**

Accept (Spotlight)

**Comment:**

This paper introduces a new social reasoning benchmark to measure the capability of LLMs, named BigToMs. The dataset is automatically generated by GPT-4 using causal templates. Then five LLMs are tested using various prompts for the BigToMs dataset. The results shows that GPT-4 demonstrates ToM capabilities that closely align with human inference patterns.

The paper is well-written and clearly explains its novel points, distinguishing it from the previous work. Although one reviewer highlighted a circularity--wherein GPT-4 both generates the data and is evaluated using it--the author adequately addressed these concerns during the reviewing process. I believe the paper has the potential to make a contribution to the research field of language and social reasoning.